# Maximizing meiotic crossover rates reveals the map of Crossover Potential

Juli Jing [1,3], Qichao Lian [1,2,3], Stephanie Durand [1] & Raphael Mercier [1] ✉

Sexual dysmorphism in the number and distribution of meiotic crossovers is seen across species but is poorly understood. Here, we disrupt multiple anti-crossover pathways in hermaphrodite Arabidopsis and analyze thousands of female and male progeny genomes. The greatest crossover increase is seen in *zyp1 recq4* mutants, with a 12-fold rise in females and 4.5-fold in males. Additional manipulation of crossover regulators does not further increase crossovers but shifts the balance between crossover pathways, suggesting competition for a shared, limited precursor pool. While wild-type crossover patterns differ between sexes, mutant crossover landscapes converge on a unique distinct profile, which we term Crossover Potential (CO$_P$). CO$_P$ can be accurately predicted using only sequence and chromatin features. We propose that CO$_P$ reflects the density of eligible recombination precursors, which is determined by genomic features and is thus identical across sexes, with sexual dimorphism resulting solely from differential regulation of their maturation into crossovers.

In sexual reproduction, meiotic recombination generates reciprocal exchanges between homologous chromosomes called crossover (CO), enhancing genetic diversity[1,2]. Notably, in the context of plant breeding, COs introduce favorable gene combinations and break up unfavorable linkages, allowing breeders to improve crop varieties. However, COs are limited in number, with typically 1-3 per chromosome per meiosis[3,4], despite a large excess of molecular precursors. CO formation is initiated by programmed DNA double-strand breaks (DSBs) which are repaired by using the homologous chromosome as a template. The number of early recombination intermediates is estimated to be 10-25 fold greater than the eventual number of COs in mammals and plants[5–9]. CO frequencies are not homogenous along chromosomes, creating a landscape of recombination with high peaks and large valleys. COs are also subjected to interference, in which one CO inhibits the formation of another nearby, effectively limiting the frequency of close double COs[10–13]. Intriguingly, CO frequency and distribution can differ markedly between females and males of the same species, a phenomenon called heterochiasmy[14]. The factors and mechanisms that shape CO distribution along the genome and differently in females and males are not well understood. In particular, it

is unclear what is the relative contribution of precursor distribution versus a differential likelihood of precursor maturation to CO is in the final landscape.

In most eukaryotes, two molecular pathways contribute to the maturation of recombination intermediates into CO, defining two classes of COs. Class I COs are promoted by the ZMMs protein, are sensitive to interference, and account for the majority of COs; class II COs are ZMM-independent and promoted by nucleases, including MUS81. Class II is a minor pathway in most eukaryotes, including plants, and is not or much less sensitive to interference[11,15].

A series of factors that limit either class I or class II COs are known in Arabidopsis. One is the dosage of the ZMM protein HEI10, whose overexpression (HEI10$^{oe}$) almost doubles the number of class I COs[16,17]. Another key factor regulating class I CO is the central part of the synaptonemal complex (SC), including the transverse filament ZYP1[18–20]. The SC forms a structure that connects the homologous chromosomes all their length during the meiotic prophase. Mutation of *ZYP1* eliminates CO interference and heterochiasmy and increases class I CO numbers ~twice[18,19], and has a cumulative effect with HEI10$^{oe}$ on the number of class I CO[21]. On the other hand, the RECQ4A and

[1]Department of Chromosome Biology, Max Planck Institute for Plant Breeding Research, Carl-von-Linné-Weg 10, Cologne, Germany. [2]Present address: Guangdong Provincial Key Laboratory for the Development Biology and Environmental Adaptation of Agricultural Organisms, College of Life Sciences, South China Agricultural University, Guangzhou 510642, China. [3]These authors contributed equally: Juli Jing, Qichao Lian. ✉e-mail: mercier@mpipz.mpg.de

RECQ4B helicases (called here RECQ4 for simplicity) redundantly prevent the formation of class II CO[3,22]. The combination of *recq4* mutation and HEI10[oe] showed an additive effect with an increase in both class I and class II COs[16], suggesting that class I and class II CO can be manipulated independently. The AAA-ATPase protein FIGL1 also limits class II COs by regulating strand invasion[23,24] with has a synergetic effect with *recq4*, reaching the largest increase of (class II) CO described to date[3,23,25,26]. Additional proteins, such as FLIP, SNI, TOP3/RMI1, HSBP, HCR1 and HCR3 have anti-CO roles in Arabidopsis, being partners or regulators of the functions described above[25,27–32]. Several combinations of anti-CO factors including higher-order ones, remained to be tested to explore the upper limit of crossover frequencies.

Here, we manipulated one, two, three, or four of these CO regulators in Arabidopsis and examined the CO frequency in thousands of progenies derived from female and male crosses. Notably, COs in *zyp1 recq4* showed a 12-fold increase in females and a 4.5-fold increase in males, accompanied by only a slight reduction in fertility. Manipulating additional CO regulators in *zyp1 recq4* did not further increase CO frequency but altered the ratio of class I and class II COs, suggesting an upper limit had been reached. The relative distributions of COs along the genome in various mutant combinations converge to a common profile in both females and males, which diverges markedly from the wild-type distribution. We termed this profile Crossover Potential (CO$_P$) and showed that it can be accurately predicted using a few sequence and chromatin features. We propose that CO$_P$ reflects the density of recombination precursors, which is determined by genomic features and is identical in both sexes, with sexual dimorphism in crossover landscapes arising solely from differences in precursor maturation into COs.

## Results

### HEI10[oe], *zyp1* and *recq4* act differently in boosting CO numbers

We analyzed the number and distribution of COs in both female and male meiosis in wild-type, single and multi-mutants, by generating biallelic- Col/L*er* F1 hybrids and reciprocally crossing them with wild-type Col (Fig. 1a, and Figure S1–S5). The obtained female and male-derived progenies (Back-cross BC1 populations) were analysed by whole genome Illumina sequencing to detect CO transmitted by female and male gametes, respectively (Fig. 1b, c). This method allows CO measurement with high accuracy, with some limitations. First, the CO position cannot be determined in regions lacking reliable markers (e.g., centromeric regions); but note that the presence/absence of CO in such areas can be reliably assessed by analyzing flanking markers. Second, with our stringent parameters, double-COs less than 90 kb apart, if they exist, and terminal CO less than 45 kb away from the telomere would be missed (see "methods"). In parallel, we counted the number of MLH1/HEI10 co-foci, which marks class I COs sites at diplotene/diakinesis[33,34], in the Col/L*er* F1 hybrids of each genotype. The MLH1/HEI10 focus counts were done in meiocytes both from ovules (female meiocytes) and anthers (male meiocytes) at the diplotene/diakinesis stage (Fig. 2a–m). At these stages, MLH1 and HEI10 foci almost perfectly co-localize (Table S1, supplementary data 2). Comparison of the number of MLH1/HEI10 foci (class I CO, counted the Col/L*er* F1 hybrids) and genetic crossovers detected by sequencing (class I + class II COs, produced in the F1 hybrids and transmitted to the BC1 populations), allows the estimation of the contribution of each pathway (Fig. 2n). Note that the mean number of CO per gamete is expected to be half the number of cytological CO (or chiasma). This is because a chiasma affects only two of the four chromatids of a chromosome pair, and a gamete inherits a single chromatid[35] (see Figure S11 in[21]). Thus, under the assumption that each MLH1 focus is converted into a CO and that all chiasma are marked by MLH1/HEI10, the frequency of genetic CO per gamete/BC1 is expected to be half of the frequency of MLH1/HEI10 foci per cell (1 focus=0.5 CO, indicated

by a dashed line in Fig. 2n). If a proportion of chiasmata/CO is nor marked by MLH1/HEI10 (i.e., class II COs), the frequency of genetic CO per gamete is expected to be higher than half of the frequency of MLH1/HEI10 foci per cell (an upward deviation from the dashed line in Fig. 2n).

In the wild type, we observed an average of 2.8 COs per female gamete, and 5.2 per male ($p < 10^{-6}$ Mann-Whitney test), confirming the previously observed heterochiasmy[36,37]. The number of MLH1 foci matches 2-fold the number of genetic COs, suggesting that in both female and male wild-type, the vast majority of crossovers are class I COs (Fig. 2n). Class II COs were suggested to represent ~15% of all meiotic COs in Arabidopsis, based on the observation of residual chiasmata in the *zmm* mutants[9]. The almost perfect match between the number of foci and the number of genetic COs that we observed suggests a more minor contribution of the class II pathway in wild-type meiosis.

In single mutants, HEI10[oe] and *zyp1*, the average CO number per transmitted gamete was substantially increased compared with the wild type, consistent with previous results[17,18]. In HEI10[oe], the CO number in females (6.2, 2.2-fold/wild type, $p < 10^{-6}$) remains lower than in males (9.9, 1.9-fold/wild type, $p < 10^{-6}$. Difference in fold increases female vs male, Mann-Whitney $p = 0.001$). In *zyp1*, the CO number is equal in female (7.1, 2.5-fold/wild type, $p < 10^{-6}$) and male meiosis (7.0, 1.3-fold/wild type, $p < 10^{-6}$. *zyp1* female vs *zyp1* male, $p = 0.41$. Differences in fold increase female vs male, $p < 10^{-6}$). In both cases, MLH1/HEI10 foci were also increased in female and male meiocytes of the F1 ($p < 10^{-4}$), matching the number of COs in their progenies (Fig. 2a, n). Thus, overexpressing HEI10 or mutating *ZYP1* exclusively increases class I COs. However, while heterochiasmy and CO interference are still present in HEI10[oe], both are abolished in *zyp1* (Fig. 1b, Figure S6).

Single *recq4* mutation increased COs to higher levels and, remarkably, inverted heterochiasmy, reaching 17.5 in females (6.3-fold/wild type, $p < 10^{-6}$) and 14 in male gametes (4.5-fold/wild type, $p < 10^{-6}$) (Fig. 1b)[3]. In contrast to HEI10[oe] and *zyp1* in which MLH1/HEI10 foci are increased, the number of MLH1/HEI10 foci was unchanged in both female or male *recq4* meiocytes compared to wild-type ($p = 0.89$ and $p = 0.09$) (Fig. 2), consistent with the role of RECQ4 in limiting specifically class II COs[22]. This is reflected in Fig. 2n, by a vertical deviation from the dashed line, with the number of genetic COs largely exceeding the expected number according to MLH1/HEI10 focus counts. Accordingly, as class II COs are not subjected to interference, CO interference is not detectable in *recq4* (Figure S6). The absence of interference is also reflected by a higher variation in the number of COs per gamete in *recq4* (and combinations, see below) than in the wild type (Fig. 2b). Thus, *recq4* boosts class II CO, with a more prominent effect in female than male meiosis.

### Maximizing CO numbers

HEI10[oe] increases class I COs while *recq4* increases class II COs. Combining *recq4* HEI10[oe], we observed an average of 23 COs in female and 17.2 in male gametes, higher than both single mutants ($p < 10^{-6}$) (Fig. 1b). This is consistent with previous studies performed in F2 population[16]. Under the hypothesis of additive effects, simply summing the wild-type counts with the gains observed in each single mutant, we would have expected 20.9 COs in female gametes [2.8 + (6.2 − 2.8) + (17.5 − 2.8)] and 18.7 in male gametes [5.2 + (9.9 − 5.2) + (14.0 − 5.2)], which is similar to the observed values. In both female and male meiosis of F1s, the numbers of MLH1/HEI10 foci in *recq4* HEI10[oe] are not modified compared to HEI10[oe] ($p = 0.47$ and $p = 0.55$) (Fig. 2a, n), suggesting that class I COs are unaffected by the *recq4* mutation. Altogether, this shows that combining *recq4* and HEI10[oe] leads to parallel increases of class I COs and class II COs provoked by respectively by HEI10[oe] and *recq4* and thus to an additive effect on the total number of COs.

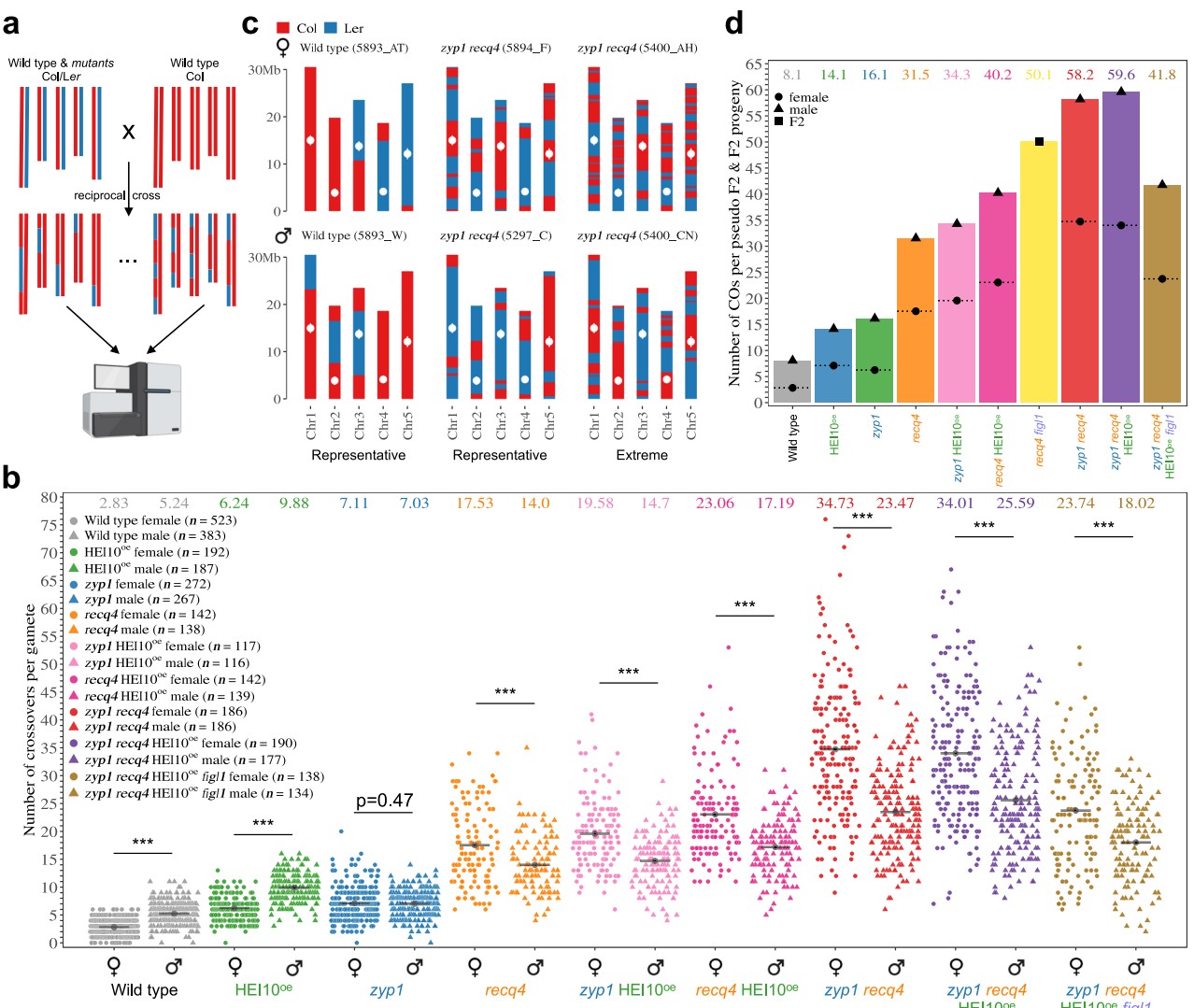

**Fig. 1 | Combination of *zyp1* and *recq4* massively increases meiotic crossover frequency. a** Schematic representation of the experimental design. Col/L*er* hybrid plants, either wild-type or carrying mutations, were crossed with wild-type Col in both directions. The resulting plant populations (BC1) were sequenced to score the COs transmitted by the female and male gametes of the hybrids. Created in BioRender. Lian, Q. (https://BioRender.com/m7vjhfg). **b** The number of COs per female and male gametes for each genotype. Each point represents the number of COs in one BC1/gamete. Circles and triangles are females and males, respectively. A black line indicates the mean and its value is specified at the top of the graph. The BC1/ gamete population size (*n*) is indicated for each genotype. Two-sided Mann-Whitney test was used to evaluate the differences in CO numbers between female and male meiosis (***: < 10⁻³). See also Figure S5 for chromosome per chromosome analysis (**c**) Representative transmitted chromatid sets are shown for selected samples. The genotype and the name of the sample are specified. Red and Blue indicate Col and L*er* genotypes, a transition marking a crossover. White points indicate centromeres. **d** The average number of COs per F2 sample (for *recq4 figl1*, yellow) or pseudo-F2 (sum of female and male averages, for all other genotypes).

Next, we combined the two factors that individually increase class I COs, HEI10oe and *zyp1*[21]. Following the same logic as above, the hypothesis of an additive effect predicts 11.7 COs in female (2.8 + 3.4 + 4.3) and 10.5 COs in male gametes (5.2 + 4.7 + 1.8). In *zyp1* HEI10oe, we observed an average of 14.7 COs in male gametes, which is higher than single mutants (*p* < 10⁻⁶) and than the additive prediction, and 19.6 in females, which is almost twice the prediction. MLH1/HEI10 foci are largely increased in F1 female and male meiosis, reaching ~29.5 in both sexes, more than each single mutant and 3-4-fold the wild-type (Fig. 2a). This indicates that HEI10oe and *zyp1* synergistically augment class I COs[21]. In male *zyp1* HEI10oe meiocytes, the number of MLH1/HEI10 foci (29.3) matches the frequency of genetic COs (Figs. 1b, 2a, n), suggesting that all COs in male *zyp1* HEI10oe are of class I COs. In contrast, in female *zyp1* HEI10oe, the MLH1/HEI10 foci counts were 29.8, which corresponds to 14.9 COs per gamete, while 19.6 COs were

observed (Figs. 1b, 2a, n). This suggests that in addition to the large boost in class I COs, class II COs are also increased in *zyp1* HEI10oe female meiosis, leading to an inversion of heterochiasmy.

In the last combination of two factors, *zyp1 recq4*, the COs number per gamete was massively increased to 34.7 in females and 23.5 in males, almost twice the number measured in *recq4*, the highest single mutant (*p* < 10⁻⁶) (Fig. 1b). This is significantly higher than the predictions under an additive hypothesis, 21.8 in females [2.8 + (7.1 − 2.8) + (17.5-2.8)] and 15.8 in males [5.2 + (7.0 − 5.2) + (14 − 5.2)], indicating a synergetic effect. The high CO levels in *zyp1 recq4* are unprecedented (compared to wild-type, 12.4-fold in females and 4.5-fold in males), being above all previously described mutants, including the champion to date *recq4 figl1* (Fig. 1b–d)[16–18,21,22,26]. The variance of CO per gamete is also large, with the extreme case of a female gamete with 76 COs distributed among the five chromosomes, while the highest number observed in a wild-type female

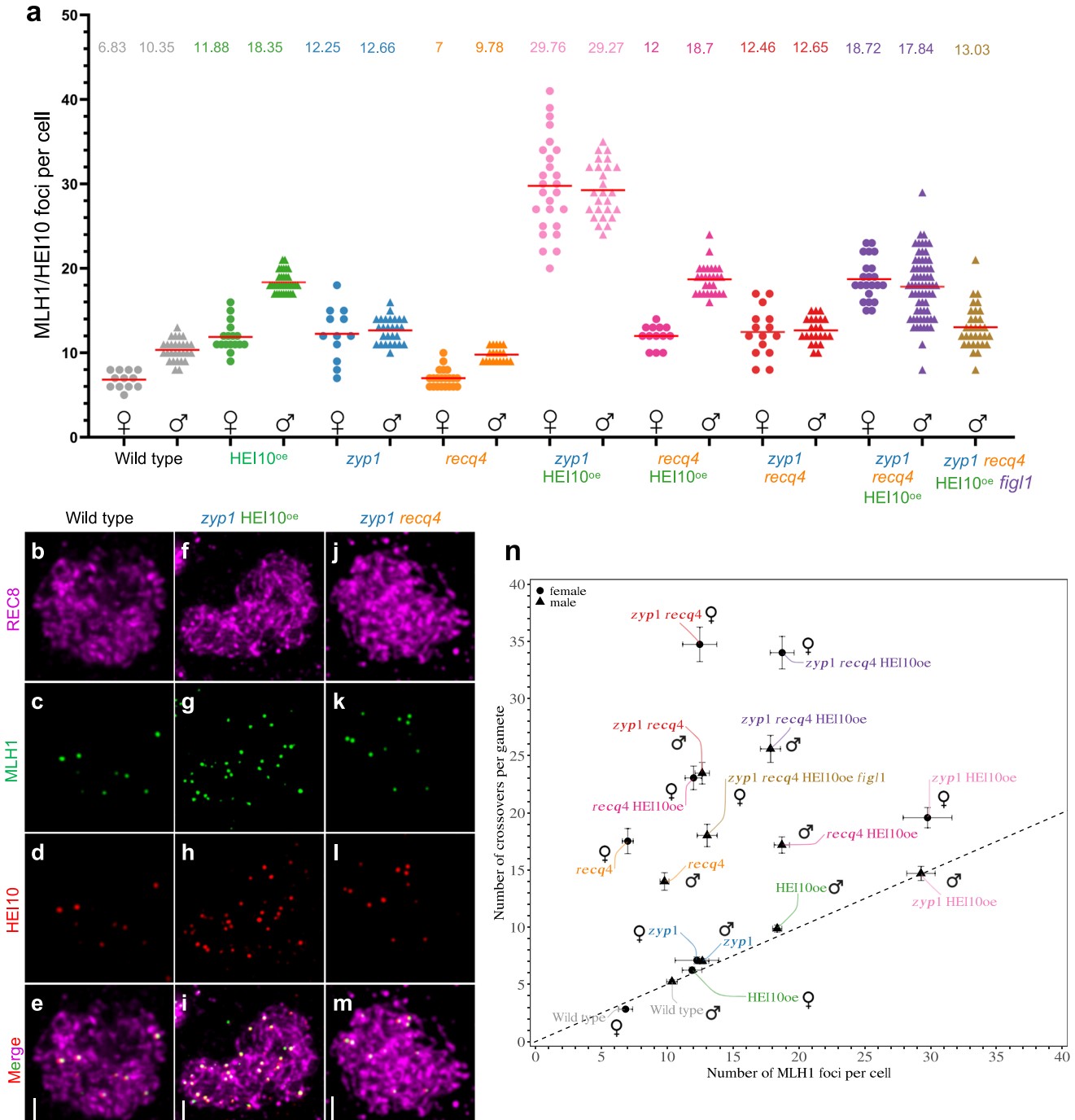

**Fig. 2 | Analysis of MLH1/HEI10 foci and contributions of crossover pathways. a** Counts of MLH1/HEI10 foci in female and male meiocytes of wild type and mutants Col/L*er* F1 hybrids. Each dot is an individual cell. Red bars and numbers indicate the mean number of MLH1/HEI10 co-foci. Circles and triangles are females and males, respectively. One or two plants were used for each genotype (see source data file for details). **b–m** Representative immunostaining of REC8 (purple), MLH1 (green) and HEI10 (red) in wild type (**b–e**), *zyp1* HEI10^oe (**f–i**) and *zyp1 recq4* (**j–m**) male diplotene meiocytes. MLH1-HEI10 co-foci indicate class I CO sites. The maximum intensity projection is shown and the complete Z-stack of the cells is available as supplementary data 2. The number of analyzed cells correspond to the number of points in Fig. 2a. Scale bar=2 μm. **n** Comparison of the numbers of genetic COs (Fig 1a) and MLH1/HEI10 foci per cell (Fig. 2a) in females and males of various genotypes. The number of samples are identical to Figs. 1a and 2a. Females are shown by circle, and males are shown by triangle. Error bars are the 90% confidence intervals of the mean. The dashed line indicates the number of genetic COs expected to be observed under the hypotheses that each MLH1/HEI10 foci is converted into a CO and all COs are marked by MLH1/HEI10 (no class II COs).

gamete is 6 COs (Fig. 1b, c). In *zyp1 recq4*, the number of MLH1/HEI10 foci (12.5 in females, 12.7 in males) are stable compared with *zyp1* (12.3 in females, 12.7 in males) (Fig. 2a, n) and do not match the number of COs, suggesting that the vast majority of CO are class II COs (Fig. 2a, n).

Mutating *RECQ4* in the *zyp1* mutant thus increases class II COs, like it does in wild-type, but with a much stronger effect (Fig. 2n). Altogether, we conclude that in *zyp1 recq4* class I COs are increased similarly to the single *zyp1* while class II COs are boosted by a synergistic effect of the

two mutations, which is particularly marked in female meiosis. This also points to a role of ZYP1 in preventing class II COs, in addition to its described role in regulating class I COs.

The results described above show that all the combinations of two mutations among HEI10[oe], *zyp1*, and *recq4* have more COs than the single mutants, predicting that the triple mutant should have even more COs. However, the CO numbers in *zyp1 recq4* HEI10[oe] did not increase significantly compared to *zyp1 recq4*, reaching 34.0 COs in females ($p = 0.68$) and 25.6 in males ($p = 0.06$), respectively (Fig. 1b), indicating that a certain upper limit might have been reached. Interestingly, the number of MLH1/HEI10 foci is increased when adding HEI10[oe] to *zyp1 recq4* in both female and male meiocytes ($p < 10^{-6}$), but not the total number of COs, suggesting that an increase in class I COs is compensated by a decrease in class II COs. This further supports the conclusion that an upper limit has been reached and that the two pathways compete for a large but limited number of CO precursors.

FIGL1 is another anti-class II CO factor that acts independently from RECQ4[3,26]. Adding *figl1* mutation to *zyp1 recq4* HEI10[oe] is thus expected to increase COs further. However, the number of COs in *zyp1 recq4* HEI10[oe] *figl1* mutant was not increased and instead was decreased compared to *zyp1 recq4* HEI10[oe] ($p < 10^{-6}$) (Fig. 1b, d). This further supports the idea of an upper limit in CO formation and shows that the *figl1* mutation can increase or decrease COs depending on the context.

### Inverted heterochiasmy in hyper-recombinant mutants

Heterochiasmy is defined as a different rate of meiotic recombination between the two sexes of the same species[38]. In wild-type Arabidopsis, the frequency of COs is higher in males (ratio female/male=0.54, $p < 10^{-6}$), with a similar ratio in HEI10[oe] (0.63, $p < 10^{-6}$). In *zyp1*, heterochiasmy is abolished with similar CO frequency in female and male gametes ($p = 0.47$) (Fig. 1b). Intriguingly, the heterochiasmy is inverted, with more COs in females than males in single *recq4* (1.25, $p = 0.00026$) and all double and triple mutants studied here: *zyp1* HEI10[oe] (1.33, $p < 10^{-6}$), *recq4* HEI10[oe] (1.34, $p < 10^{-6}$), *zyp1 recq4* (1.48, $p < 10^{-6}$), *zyp1 recq4* HEI10[oe] (1.33, $p < 10^{-6}$), and *zyp1 recq4* HEI10[oe] *figl1* (1.32, $p = 1.02 \cdot 10^{-6}$) (Fig. 1b). This shows that while CO levels are higher in males than in females in wild-type, the potential for CO formation is higher in females.

### Fertility and meiosis are only marginally affected in *zyp1 recq4*

In most eukaryotes, CO frequencies are limited to 1-3 per chromosome, leading to the idea that a higher frequency of COs could be detrimental, possibly disturbing chromosome segregation during meiotic divisions and fertility. We estimated the fertility by counting the number of seeds per fruit (silique). Different mutants showed varying degrees of reduction in fertility, but the fertility is poorly correlated with CO frequency (Fig. 1b, Fig. 3a). The fertility of *recq4* was not significantly reduced compared to the wild type, despite a massive increase in COs (6.3-fold in females, 2.7-fold in males), consistent with previous results[3,16] (Fisher's LSD test, $p > 0.9999$, Fig. 1b, Fig. 3a). In *zyp1*, fertility is modestly but significantly affected ($p < 0.0001$, Fig. 3a), probably because of the failure to ensure the obligate CO[18,19]. Strikingly, *zyp1 recq4* showed a fertility similar to *zyp1* ($p = 0.9913$, Fig. 3a, b), showing that the dramatic increase in COs is not associated with a further reduction in fertility. Despite a ~12-fold elevated CO frequency in female meiosis, the fertility of *zyp1 recq4* is still ~75% of wild-type (Fig. 3a, b). Further, analyses based on genome coverage by sequencing did not detect any aneuploids in the progeny of *zyp1 recq4* ($n = 2 \times 186$) suggesting that chromosome missegregation at meiosis is absent or rare (Fig. 3c). However, chromosome spreads of male meiocytes revealed abnormal chromosome structures in ~16% of metaphase I cells, suggestive of abnormal recombination intermediates, which might cause reduced fertility in *zyp1 recq4* (Fig. 3e, f, k, l, p).

HEI10[oe] showed a significant fertility reduction ($p < 0.0001$, Fig. 3a), consistent with previous studies[16], possibly due to a chromosomal rearrangement associated with the transgene in the HEI10[oe] C2 line[17], which is heterozygous in the hybrid context. Consistently chromosome connections were observed in 34% of HEI10[oe] meiotic cells (Fig. 3g, m, p). Aneuploids were also detected in the progeny of all genotypes with HEI10[oe] (Fig. 3c) whereas no aneuploids were detected in *recq4* and *zyp1 recq4* progenies (Fig. 3c). Fertility of *zyp1 recq4* HEI10[oe] *figl1* was not reduced compared with *zyp1 recq4* HEI10[oe] ($p = 0.9961$, Fig. 3a, b), suggesting that plants can still maintain a certain level of fertility even with disturbed strand-invasion and high recombination.

### Meiotic chromosome fragmentation and reduced fertility in *zyp1 mus81*

As shown above, mutating *ZYP1* in certain contexts (*recq4*, female HEI10[oe]) provokes an increase in COs but not MLH1/HEI10 foci, suggesting an anti-class II CO function for ZYP1, in addition to its established role in regulating class I COs. Mutating the nuclease MUS81 in anti-class II factors in Arabidopsis provoked meiotic catastrophe[22,23,39,40], presumably because DNA joint molecules remain unrepaired. Similarly, we observed some chromosome fragments in 40% of anaphase I meiotic cells in *zyp1 mus81* (Figure S8). In addition, fertility in *zyp1 mus81* was reduced by 25% compared to wild type or *mus81* (Figure S8j). This suggests that ZYP1 prevents the formation of joint molecules that need MUS81 to be repaired. However, this role is likely minor or redundant as no increase of class II COs is observed in the single *zyp1* mutant (Fig. 2n)[18].

### Revealing the distribution of Crossover Potential

Next, we examine the distribution of COs along the chromosomes to test whether or not the increase observed in the different mutants is homogeneous (Fig. 4). In all the mutant combinations with large CO increases, COs tend to accumulate in distal regions in both females and males. It is especially striking when looking at the highest recombining genotypes *zyp1 recq4* and *zyp1 recq4* HEI10[oe] (Fig. 4a, b). The same is observed when merging the female and male data (pseudo-F2), and in the previously analyzed *recq4 figl1* F2 populations (Fig. 4c)[26]. Intriguingly, the crossover landscapes in the different hyper-recombining mutants were highly consistent with each other, with common peaks and valleys (see also below). Looking at the local fold increase compared to the wild type (Figure S9), we detected common hotspots of CO increase in mutants, some above 200-fold in *zyp1 recq4* and *zyp1 recq4* HEI10[oe], notably in distal regions in females where CO frequency is low in wild-type and high in the mutants. In contrast, regions in the periphery of the centromeres where CO frequencies are relatively high in wild-type meiosis, correspond to recalcitrant zones with no or limited increase in the mutants.

Zooming on closer centromere-proximal region, we examined the Non-Recombining Zones (NRZs) previously defined as the interval spanning the centromere with a complete absence of CO in the wild type, as scored in 3,613 gametes containing a total of 14,397 COs[41] (Fig. 4d). Strikingly, with such an unprecedented increase in CO numbers in the mutants described here, the NRZs persisted in resisting recombination and barely had any CO, despite our dataset containing a total of 70,022 COs. No COs were detected in NRZs in *zyp1 recq4* (10,825 COs), and only one CO was found in each population of *recq4* (3,675 COs), *recq4 figl1* (15,543 COs), and *zyp1 recq4* HEI10[oe] (10,991 COs) (Fig. 4d). This strongly suggests that other mechanisms, that do not depend on HEI10, ZYP1, RECQ4 or FIGL1, prevent CO formation in the NRZs.

Crossover landscapes in various mutants appear to have similar landscapes (Figs. 4, 5). To explore this formally, and compare the shape of the distributions independently of the absolute frequencies, we normalized the data by calculating the distribution of relative

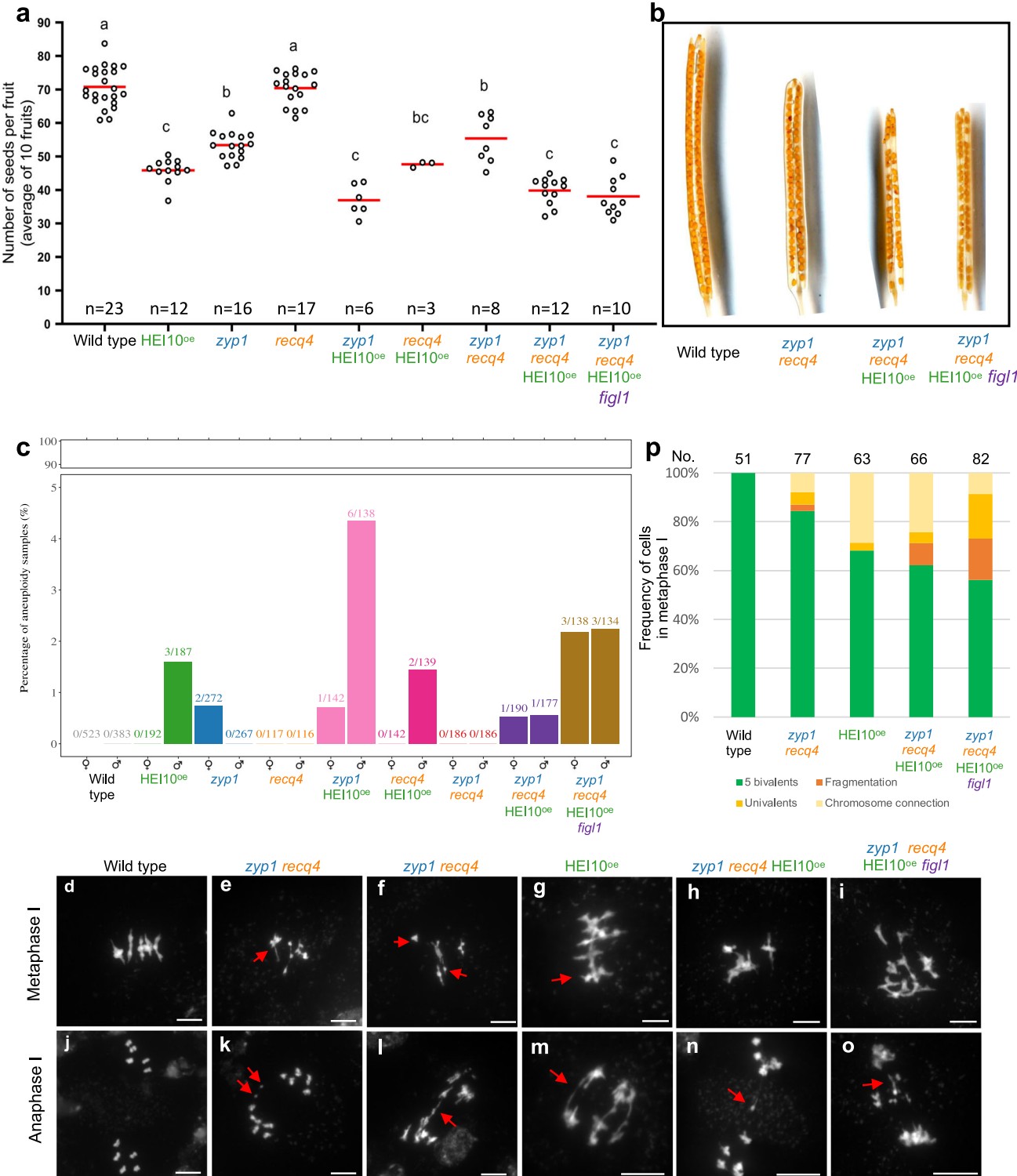

Fig. 3 | **Analysis of fertility and meiosis defect. a** Quantification of fertility. Each dot represents the fertility of an individual plant, measured as the number of seeds per fruit averaged on ten fruits. The red bar shows the mean. All plants were grown in parallel, and the wild-type controls were siblings of the mutants. The number n of analyzed plants is indicated. Letters (**a**–**c**) above each data point group indicate statistically significant differences among genotypes, as determined by one-way ANOVA followed by Tukey's multiple comparisons test ($p < 0.05$). Genotypes sharing the same letter are not significantly different from one another. **b** Representative cleared fruits of wild type, *zyp1 recq4*, *zyp1 recq4* HEI10^oe and *zyp1 recq4* HEI10^oe *figl1* mutants in Col/L*er* background. **c** The percentage of aneuploid samples detected in each population. The proportion of aneuploid samples in each population is shown on top of the bars. **d**–**o** DAPI-stained meiotic chromosome spreads from Col/L*er* male meiocytes in wild type (**d, j**), *zyp1 req4* (**e, f, k, i**) HEI10^oe (**g, m**), *zyp1 recq4* HEI10^oe (**h, n**) and *zyp1 recq4* HEI10^oe *figl1* (**i, o**). **d**–**i** Metaphase I. **j**–**o** Anaphase I. Scale bar = 10 μm. Red arrows pointed out abnormal chromosome connections, fragments and chromosome threads. **p** Quantification of different chromosome behaviors at metaphase I in wild type, *zyp1 recq4*, HEI10^oe, *zyp1 recq4* HEI10^oe and *zyp1 recq4* HEI10^oe *figl1*. Cells were categorized according to normal (5 bivalents) and abnormal chromosome behavior (fragmentation, univalent and chromosome connection). The number of analyzed cells is indicated above the bar.

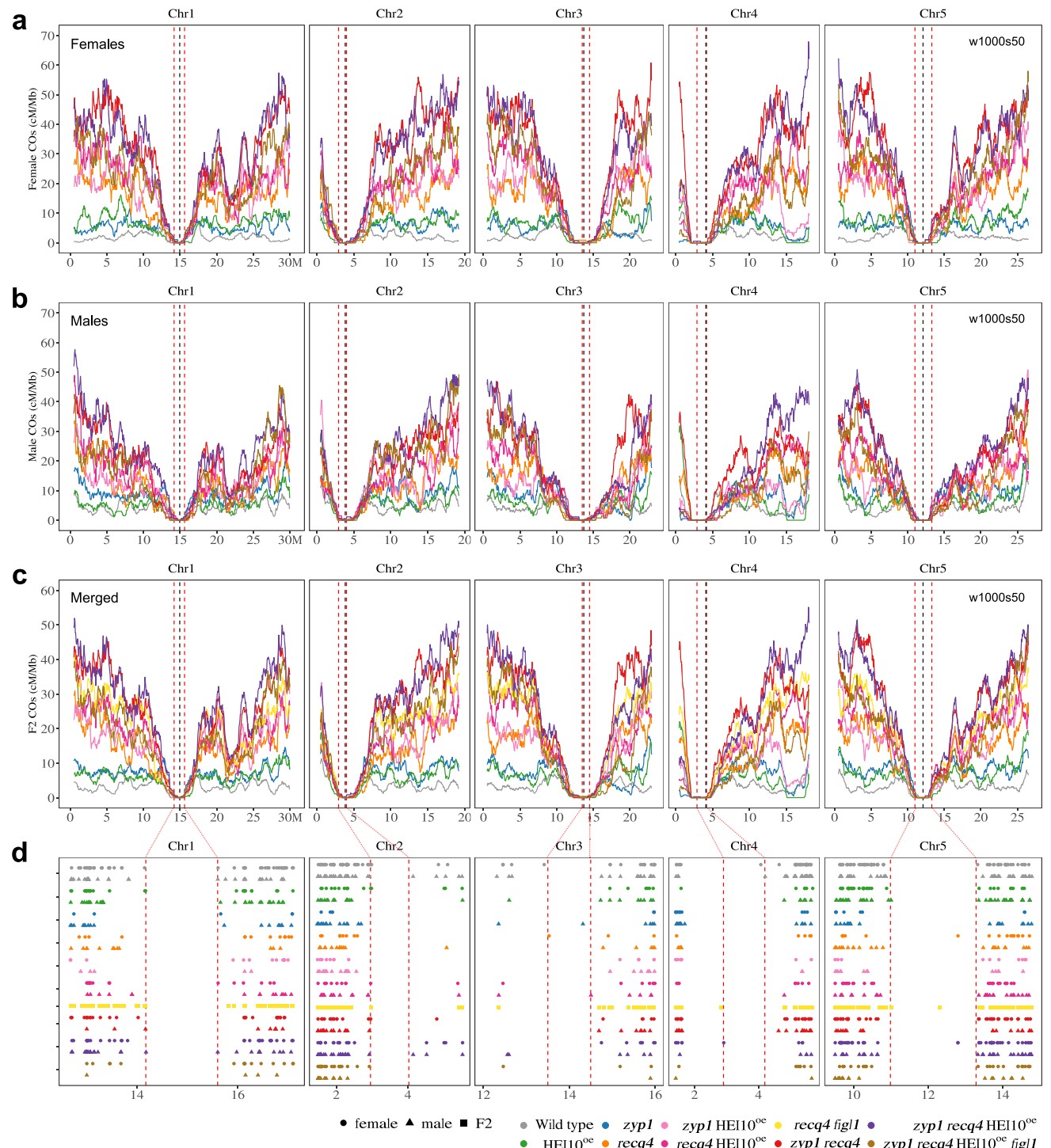

**Fig. 4 | Chromosomal distribution of COs in female, male, and F2 contexts.** **a**, **b** The distribution (sliding window-based, window size 1 Mb, step size 50 kb) of COs along chromosomes in female (**a**) and male (**b**) of wild type, HEI10^oe, *zyp1*, *recq4*, *zyp1* HEI10^oe, *recq4* HEI10^oe, *zyp1 recq4*, *zyp1 recq4* HEI10^oe and *zyp1 recq4* HEI10^oe *figl1*. **c** The distribution (sliding window-based, window size 1 Mb, step size 50 kb) of COs along chromosomes in F2 or pseudo F2 of wild type, HEI10^oe, *zyp1*, *recq4*, *zyp1* HEI10^oe, *recq4* HEI10^oe, *recq4 figl1*, *zyp1 recq4*, *zyp1 recq4* HEI10^oe and *zyp1 recq4* HEI10^oe *figl1*. **d** The zoom of the CO position in the centromere proximal regions (Non-Recombining Zones, NRZs). Each point is a CO, circles, triangles and squares are females, males and F2s, respectively. The vertical dashed lines in red indicates the position of marker COs of NRZs, lines in black shows the middle position of centromeres.

frequency per chromosome (i.e., what proportion of COs that occurred in this chromosome occurred in a given interval) (Fig. 5 and Figure S10–12). In wild-type females, the density of COs was the highest in the periphery of centromeres, and low at chromosome ends (Fig. 5a).

In the wild-type males, COs are also frequent in the periphery of centromeres, but chromosome ends are also CO-rich (Fig. 5b). Consequently, female and male distributions are poorly correlated in the wild type (Spearman's correlation *r* = 0.45, Fig. 5h). Both female and

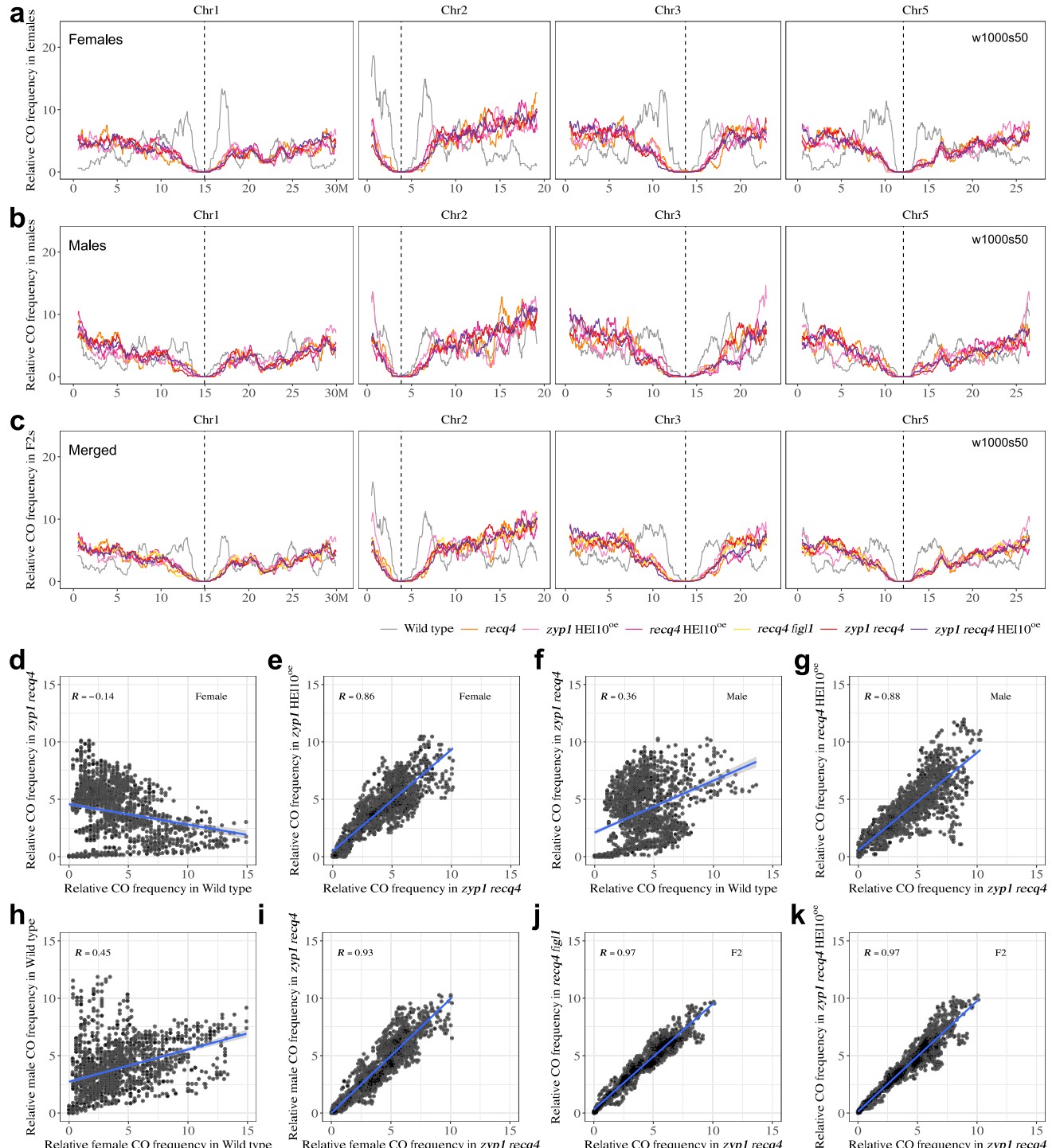

**Fig. 5 | Comparison of the genome-wide CO landscape in female, male, and F2 contexts. a, b** The distribution (scaled per chromosome, sliding window-based, window size 1 Mb, step size 50 kb) of relative CO frequency along chromosomes in female (**a**) and male (**b**) of wild type, *recq4*, *zyp1* HEI10^oe, *recq4* HEI10^oe, *zyp1 recq4* and *zyp1 recq4* HEI10^oe. **c** The distribution (scaled per chromosome, sliding window-based, window size 1 Mb, step size 50 kb) of COs along chromosomes in F2 or pseudo F2 of wild type, *recq4*, *zyp1* HEI10^oe, *recq4* HEI10^oe, *recq4 figl1*, *zyp1 recq4* and *zyp1 recq4* HEI10^oe. **d–k** Spearman's correlation tests of relative CO frequencies (each point in an interval, as defined as Fig. 5a–c) comparing between genotypes and sexes. The blue line was added by using "lm" method, with 0.95 confidence intervals marked by gray shading.

male *zyp1 recq4* distributions differ markedly from the wild-type distributions ($R = -0.14$ and $R = 0.36$, Fig. 5a–b, d, f), but are strikingly similar to each other ($r = 0.93$, Fig. 5a, b, i). Further, the CO distributions in various mutants were similar to each other, but distinct from the wild-type, in both females (e.g., $r = 0.86$, *zyp1* HEI10^oe vs *zyp1 recq4*; Fig. 5a, d–e, Figure S10) and males (e.g., $r = 0.88$, *recq4* HEI10^oe vs *zyp1*

*recq4*; Fig. 5b, f, g, Figure S11). The CO landscape in the F2 populations of *recq4 figl1* and *zyp1 recq4* are also strikingly consistent ($r = 0.97$, *recq4 figl1* vs *zyp1 recq4*; r = 0.97, *zyp1 recq4* HEI10^oe vs *zyp1 recq4*; Fig. 5c, j, k, and Figure S12)[26]. CO landscapes in *recq4*, *zyp1* HEI10^oe, *recq4* HE10^oe, *recq4 figl1*, *zyp1 recq4*, *zyp1 recq4* HEI10^oe are all very similar, with correlations in the range 0.75–0.92 (Figure S12).

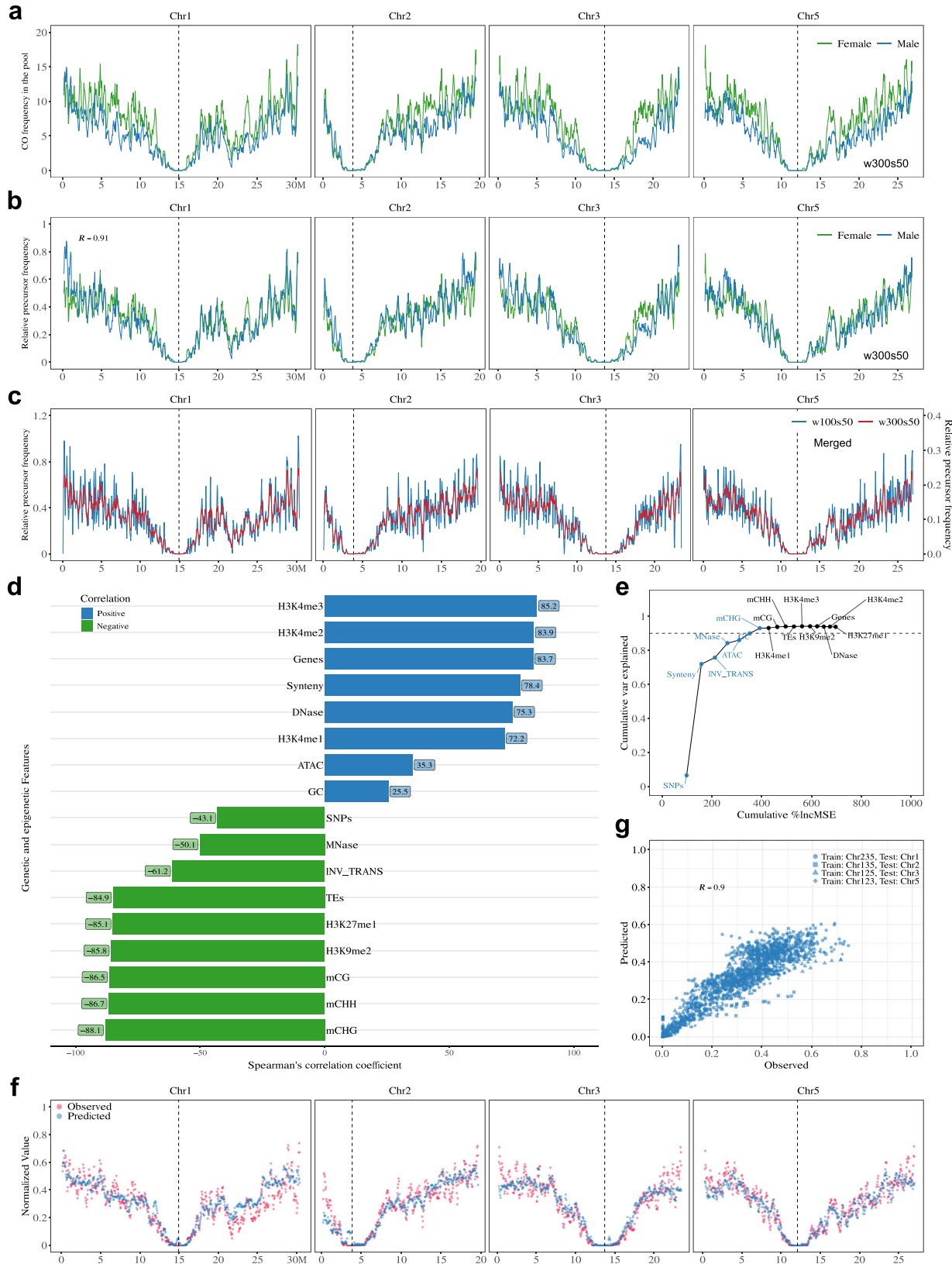

The convergence towards a similar pattern in females and males in different mutants suggests the existence of a common underlying feature. We propose that this distribution of Crossover Potential (CO_P) represents the distribution of eligible recombination intermediates (i.e., DSBs that are being repaired with the homologous chromosomes, potentially leading to a CO). We then combined the 21,104 female and 15,188 male COs from the various mutant combinations (*recq4, zyp1* HEI10^oe, *recq4* HEI10^oe, *zyp1 recq4, zyp1 recq4* HEI10^oe, and *zyp1 recq4* HEI10^oe *figl1*) to create high-resolution maps of CO_P in females and males, separately (Fig. 6a). The female and male CO_P maps are strikingly similar, the female curve being slightly above the male one. When normalized, the two CO_P curves almost perfectly overlap (Fig. 6b,

**Fig. 6 | Association and prediction of precursor distribution with genetic and epigenetic features. a** The distribution (sliding window-based, window size 300 kb, step size 50 kb) of CO frequency along chromosomes in females and males, respectively, by merging *recq4*, *zyp1* HEI10[oe], *recq4* HEI10[oe], *zyp1 recq4*, *zyp1 recq4* HEI10[oe] and *zyp1 recq4* HEI10[oe] *figl1*. **b** The distribution (scaled per genome, sliding window-based, window size 300 kb, step size 50 kb) of relative precursor frequency along chromosomes in females and males, respectively, by merging *recq4*, *zyp1* HEI10[oe], *recq4* HEI10[oe], *zyp1 recq4*, *zyp1 recq4* HEI10[oe] and *zyp1 recq4* HEI10[oe] *figl1*. **c** The distribution (scaled per genome, sliding window-based, window size 100 kb and 300 kb, step size 50 kb) of relative precursor frequency along chromosomes in F2s, by merging females and males of *recq4*, *zyp1* HEI10[oe], *recq4* HEI10[oe], *zyp1 recq4*, *zyp1 recq4* HEI10[oe] and *zyp1 recq4* HEI10[oe] *figl1*, and F2s of *recq4 figl1*. The vertical dashed lines in black shows the middle position of centromeres. **d** Spearman's correlation test shows the comparison with features along chromosomes, with differences in color and length according to the correlation scale. SNPs (SNPs density between Col and Ler), INV_TRANS (inversions and translocations between Col and Ler), Synteny (collinearity between Col and Ler), Genes, TEs and GC (expressed gene in meiocytes, TE and GC density), ATAC and DNase (chromatin accessibility, ATAC-seq and DNase-seq, log2(Tn5/gDNA) and log2(DNase/gDNA), in floral tissues), H3K4me1/2/3, H3K9me2, H3K27me1 (euchromatin, heterochromatin, and Polycomb histone marks, ChIP-seq, log2(ChIP/input), in flower buds), mCG, mCHG and mCHH (DNA methylation in CG, CHG, and CHH contexts, proportion methylated cytosine, in male meiocytes), MNase (nucleosome occupancy, MNase-seq, log2(MNase/gDNA), in buds). **e** The cumulated proportion of variation that can be explained with the features at the genome scale. The top seven most important features are colored, for which the cumulative proportion of variation that can be explained reaches the plateau. **f** The chromosomal distribution of observed and predicted precursor maps. The precursor profiles of individual chromosomes were predicted using profiles of the top seven most important features from the other three chromosomes. **g** The Spearman's correlation test between the predicted and observed precursor distributions. The training-testing dataset is differentiated in shapes.

Spearman's correlation r = 0.91), showing that the distribution of $CO_P$ is shared between the two sexes. These features of $CO_P$ strikingly deviate from those of wild-type COs, which are more frequent in males than females and very differently distributed. We then combined the female and male COs of the mutants listed above, together with the *recq4 figl1* F2 population, totaling 49,482 COs, and generated a high-resolution universal $CO_P$ map (Fig. 6c). The $CO_P$ is not homogenous along chromosomes with null values in the centromeric regions and globally higher frequencies toward chromosome ends, and sharp peaks and valleys.

### Genomic features can predict the CO potential

We next wondered what could shape the $CO_P$ landscape. We performed correlation analysis with 17 different genomic features (Fig. 6d), including GC content, meiotically expressed genes[42], transposable elements, sequence divergence between the two parental lines, chromatin accessibility, euchromatic and heterochromatic histone modification marks, DNA methylation, and nucleosome occupancy[36]. The distribution of $CO_P$ is strongly positively correlated with euchromatic histone modification markers (H3K4me3, r = 0.85), density of expressed genes in meiocytes (r = 0.84), open chromatin (DNase, r = 0.75), and synteny (r = 0.78). Oppositely, significant negative correlations were found with DNA methylation (mCHG, r = −0.88), heterochromatic histone modification marks (H3K9me2, r = −0.86), density of TEs (r = −0.85) and inversions and translocations (r = −0.61). This result suggests that $CO_P$ is favored by certain open chromatin states and disfavored by sequence divergence.

We then employed a machine-learning algorithm (random forest) to measure the capacity of the 17 features to predict the $CO_P$ landscape. Overall, 94% of the genome-wide variation can be explained by a model developed with all 17 features (Fig. 6e). Among the 17 features, SNP density was assessed as the most predictive, which alone explained 6.4% of the genome-wide variation (Fig. 6e) and 28.4% of the variation along chromosome arms (Figure S13b). By adding features step by step in the order of importance, we found that the top seven and eight features can explain 93% of the genome-wide variation and 90.6% of the variation along chromosome arms, respectively (Fig. 6d, Figure S13b). To explore the predictive performance in the $CO_P$ landscape of the model, we next performed 4 times cross-validation by using three chromosomes as the training dataset and one chromosome as the testing dataset (Fig. 6f, g, Figure S13c, d). Considering the top seven and eight features for genome-wide and chromosome arms, the model trained from the training dataset worked well with the testing dataset, resulting in a strong correlation (r = 0.9, Fig. 6f, g, and r = 0.86, Figure S11c, d) between the prediction and observation. The capacity of these features to accurately predict the landscape suggests that the $CO_P$ might be entirely determined by (epi)genomic characteristics, including chromatin state and sequence divergence.

## Discussion

Meiotic crossover frequencies are naturally limited to a few per chromosome. Here, we have shown an unprecedented elevation of meiotic crossovers in Arabidopsis by simultaneously mutating the synaptonemal complex ZYP1 and the anti-recombination helicase RECQ4. We observed an average of 34.7 and 23.5 COs per female and male *zyp1 recq4* gametes, respectively. This corresponds to 58 COs per generation in *zyp1 recq4*, to be compared with 8.1 in wild-type and 50 in the previous champion *recq4 figl1*[3,26]. The fertility is only moderately affected in *zyp1 recq4* and no genomic instability was observed, opening a possibility to manipulate recombination for the benefit of plant breeding. Higher CO levels could be leveraged to enhance genetic mixing of genetic information in the early steps of breeding, reduce the size of introgression to elite lines, and facilitate the identification of genetic determinants of traits[26]. The mutation of *RECQ4* increases CO numbers in rice, tomato, and pea[43,44], suggesting that manipulation of this factor can be useful in a wide variety of crops. However, it should be noted that the mutation of some anti-crossover factors, such as *FIGL1*, can have little effect on fertility in Arabidopsis and a much more dramatic effect in crop species[44]. Similarly, the *recq4* mutation does not reduce fertility in Arabidopsis, rice, and tomato but does affect fertility in peas and wheat, which may limit its usefulness[44,45]. Mutation of *ZYP1* in rice also increased class I CO number, but a *ZYP1* RNAi reduced them in Barley[46,47], calling for testing the effects of *zyp1*, and *zyp1 recq4* in more species.

Mutating both ZYP1 and RECQ4 provoked a remarkable elevation in CO numbers, beyond what was expected under the hypothesis of simple additive effects of both mutations on class I and class II COs, respectively. These massive increases correspond to class II COs, as indicated by MLH1/HEI10 not marking extra COs. This points to a function of ZYP1 in preventing class II COs, which is revealed only when RECQ4 is absent, in addition to its documented function in regulating class I COs[18,19]. It suggests that recombination precursors (DNA double-strand breaks) could be increased in the absence of ZYP1, as shown when the homologous protein Zip1 is defective in yeast[48,49]. If the same scenario happens in *zyp1 recq4*, the additional DSBs would produce additional recombination intermediates, that would be repaired as non-crossover in the presence of RECQ4, but converted into class II COs in its absence. Another non-exclusive possibility is a role of the synaptonemal complex in preventing the formation of aberrant recombination intermediates (e.g., Multiple invasions from a single DSB). This is supported by the slight chromosome fragmentation observed in *zyp1 mus81*, pointing to the role of the nuclease MUS81 in repairing recombination intermediates in the absence of ZYP1. The more extensive fragmentation observed in rice *zyp1 mus81*[50] suggests that this function could be more prominent in this species. Finally, one may speculate a role of ZYP1 in channeling the repair of a proportion of DSBs to the sister chromatid, leading to an increase of

inter-homolog intermediate in *zyp1* that would synergize with the anti-CO effect of RECQ4.

Combining HEI10[oe], *zyp1* or *recq4* two by two systematically increased CO beyond the level of the single mutants. This predicted that combining the three should increase even further CO frequencies. However, adding HEI10[oe] to *zyp1 recq4*, the double mutant with the strongest effect, did not further increase COs. Interestingly, HEI10[oe] did increase the number of MLH1/HEI10 foci in both sexes in *zyp1 recq4*, without changing the total number of COs. This suggests that over-expressing HEI10 does, as expected, increase the number of class I COs, but that this is at the expense of class II COs. This suggests that a maximum number of COs has been reached and that at these high levels, the class I and class II pathways compete for a limited number of precursors. Intriguingly, while *figl1* can enhance CO frequencies in *recq4*[3,26], adding the *figl1* mutation to *zyp1 recq4* HEI10[oe] does not increase COs, but decreases the CO frequency. We suspect that in this context, aberrant recombination molecules are formed, limiting their possibility to be matured into COs. In any case, it appears that the CO frequency observed in *zyp1 recq4* corresponds to some kind of maximum. This maximum corresponds to ~70 chiasmata/COs per female meiocyte (average of 35 COs per gamete) and ~48 chiasmata/COs per male meiocyte (average of 24 COs per gamete). It is possible that this does not correspond to an absolute maximum, and playing with additional factors, for example, related to chromatin state or mismatch detection, may further enhance CO frequencies[51,52]. Based on DMC1/RAD51 foci, it is estimated that ~200 DSBs are produced per male meiocyte[53]. If we consider that 1/3 are repaired on the sister (absence of bias, one sister chromatid and two homologous chromatids)[54], and that the class II pathway is not biased in the repair of CO precursors (1:1 CO/NCO), we would expect ~66 COs, to compare with the estimated ~48 chiasmata in *zyp1 recq4* males. If all the considerations are correct, this suggests that additional mechanisms prevent DSB maturation into COs. Intriguingly, we observed an inversion of heterochiasmy in all mutants in which class II COs are deregulated (all the genotypes with the *recq4* mutation, and female *zyp1* HE10[oe]), with more COs in females than in males. One parsimonious hypothesis is that in all contexts, including wild-type, more DSBs are formed in females than males making female meiosis in mutants more prone to class II COs, but that the CO designation process in wild-type which is linked to the length of the SC and impassive to DSB counts would lead to fewer COs in females than males.

Strikingly, in female and male meiosis of various mutants, the CO distribution converged to a common profile, leading us to propose the concept of Crossover Potential (CO$_P$), whose density map is revealed when the CO designation process is deregulated. In this view, the CO$_P$ determines where COs might occur, combining the local capacity to experience DNA double-strand breaks and the availability of a viable repair template on the homologous chromosome. Accordingly, open chromatin markers (positively correlated) and sequence divergences (negatively correlated) can together efficiently predict CO$_P$. It would be interesting to establish the CO$_P$ in other species with different (epi)genomic characteristics compared to Arabidopsis (e.g., higher transposon or polymorphism density) and to test if similar features can predict it. CO$_P$ would be responsible for the local placement of COs in genes observed in many species[55–59]. Note that PRDM9, which drives DSB positions in many mammals, dramatically affects CO$_P$ distribution[60]. Importantly, CO$_P$ very partially dictates the final CO distribution because of the CO designation process[61,62] that determines the fate of eligible recombination intermediates. A striking example is the sexual dimorphism in the hermaphrodite Arabidopsis, where female and male CO landscapes markedly differ despite an identical genome and CO$_P$ map. The limited influence of CO$_P$ on the global CO distribution explains why the megabase-scale CO landscape is largely independent of sequence divergence[36]. While CO$_P$ appears to be determined at the (epi)genomic level (chromatin state and sequence divergence), the CO designation process is a chromosomal

event influenced by the higher-order spatial organization in the synaptonemal complex[21,63–65]. The CO distribution is thus determined by the combination of the CO$_P$, which is particularly important at the local scale and can dismiss some regions, and the designation process that shapes the global chromosomal landscape and dictates CO counts and heterochiasmy.

## Methods

### Plant materials and growth conditions

*Arabidopsis thaliana* plants were cultivated in Polyklima growth chambers (16-h day, 21.5 °C, 280 μM; 8-h night, 18 °C: 60% humidity). The following Arabidopsis lines were used in this study: Wild type Col-0(186AV1B4) and L*er* − 1 (213AV1B1) from the Versailles stock center (http://publiclines.versailles.inra.fr/). This study used *zyp1-1* in Col and *zyp1-6* in L*er*[18], *recq4a* in Col (*recq4a-4*, N419423)[66], *recq4a-W387X* in L*er*[22], *recq4b* in Col (*recq4b-2*, N511330)[66], *figl1-19* in Col (*SALK_089653*), *figl1-12* in L*er*[23] and *mus81-2* in Col[67]. The HEI10 over-expression line is Col HEI10 line C2[17]. Genotyping of the mutants was carried out by PCR amplification (Supplementary Table 2).

### Generation of multi-mutants

The combinations of different mutants (Col/L*er* hybrids) were obtained by a series of crossing schemes (Table S2 and Figs. S1–4). *zyp1* HEI10[oe] was produced previously[21]. These eight different mutants and their sister wild-type control plants were backcrossed with Col plants as males and females to get the BC1 populations for sequencing and subsequent CO analysis. BC1 plant populations were grown for three weeks (16-h day/ 8-h night) and four days in the dark. 100–150 mg leaf samples from individual plantlets were collected from the BC1 populations[68]. Each plantlet was used as an individual sample for DNA extraction, library preparation and sequencing. The complete list of samples is provided in the Source data file of Fig. 1.

### CO identification and analysis

In this study, the female and male population of wild type (95 and 89 plants), HEI10[oe] (48 and 46 plants), *zyp1* (42 males), *recq4* HEI10[oe] (142 and 139 plants), *zyp1 recq4* (186 and 186 plants), *zyp1 recq4* HEI10[oe] (190 and 177 plants), *zyp1 recq4* HEI10[oe] *figl1* (138 and 134 plants), were sequenced by Illumina HiSeq3000 (2 × 150 bp) conducted by the Max Planck-Genome-center (https://mpgc.mpipz.mpg.de/home/). The raw sequencing data of the female and male population of wild type (428 and 294 plants, ArrayExpress number: E-MTAB-11254[36]), HEI10[oe] (144 and 141 plants, ArrayExpress number: E-MTAB-11696[21] and *zyp1* (272 and 225 plants, ArrayExpress number: E-MTAB-9593[18], E-MTAB-11696[21]) from previous studies were also included in this study. In total, we analyzed female and male populations of 523 and 383 wild type, 192 and 187 HEI10[oe], 272 and 267 *zyp1*, 142 and 138 *recq4* HEI10[oe], 142 and 139 *recq4* HEI10[oe], 186 and 186 *zyp1 recq4*, 190 and 177 *zyp1 recq4* HEI10[oe], 138 and 134 *zyp1 recq4* HEI10[oe] *figl1* plants. The number of COs, sequencing depth and source of each individual sample is provided in the Source data file of Fig. 1.

The raw sequencing data were quality-controlled using FastQC v0.11.9 (http://www.bioinformatics.babraham.ac.uk/projects/fastqc/). The sequencing reads were aligned to the *Arabidopsis thaliana* Col-0 TAIR10 reference genome[69,70], using BWA v0.7.15-r1140[71], with default parameters. A set of Sambamba v0.6.8[72] commands was used for sorting and removing duplicated mapped reads. The creation of the high-confidence SNP marker list between Col and L*er*, meiotic CO detection (a sliding window-based method, with a window size of 30 kb and a step size of 15 kb), A double-CO must be supported by at least five windows (total of 90 kb), and a terminal CO by at least two windows (45 kb). Check and filtering of low covered and potential contaminated samples were performed according to previous protocols[18,21,36,73,74]. Samples of each population were randomly selected to check predicted COs manually by inGAP-family v1.0[74].

The Coefficient of Coincidence (CoC) was calculated using MADpattern v1.1[64,75], with 10 intervals. The chromosome 4 was excluded from interference analyses because of a translocation associated with the HEI10 transgene[16] and potential inversion in our L*er* line[18].

For CO distribution analysis, we refined the position of the marker COs of each NRZ boundary against the Col-TAIR10 reference genome used in this study. In addition, it should be noted that a translocation associated with the HEI10 C2 transgene at the short arm and a megabase-scale inversion at the long arm of chromosome 4, which may introduce bias of the CO distribution and thus chromosome 4 was excluded[16].

To define hot and cold zones of CO burst from hyper recombination mutants, we examined COs in 1-Mb windows with 50-kb sliding and defined (i) the hot zones as common regions in *zyp1* HEI10[oe] and at least 3 other mutants, with at least a two-fold increase than the median compared with wild-type, and (ii) the cold zones as common regions in *zyp1* HEI10[oe] and at least 3 other mutants, with at most a half-fold increase than the median compared with wild-type.

For the genome-wide CO potential profile analysis by machine-learning algorithm, the chromosomal profiles of CO potential, genomic and epigenomic features were estimated in 300-kb windows with 50-kb steps along chromosomes. Then, all the random forest models were trained using randomForest v4.6-14 package in R, with the setting of "mtry=3, importance=TRUE, na.action=na.omit, ntree=2000".

### Aneuploidy screening by whole−genome sequencing

The genome was first cut into non-overlapping 100 kb windows, whose sequencing depth was estimated by Mosdepth v0.2.7[76] with parameters of "-n –fast-mode –by 100000". Then, pairwise testing of sequencing depths between chromosomes of individuals was performed using the Mann−Whitney test, and the p values were adjusted using the fdr method. A pair of tested chromosomes with fold change > 1.2 and p value < 1e − 20 was considered aneuploid.

### Meiocyte RNA-seq analysis

The RNA-seq dataset of *Arabidopsis thaliana* meiocytes from a previous study[42] was downloaded from the NCBI SRA database (SRR5209212 and SRR5209213). First, quality checking of the raw sequencing reads was performed by using FastQC. Then, HISAT2 v2.1.0[77] was used to align the reads against the Col-0 TAIR10 reference genome. Gene expression was normalized as TPM, which was calculated by StringTie v2.0.6[78] with default parameters.

### Cytology and image processing

Meiotic chromosome spreads were performed as previously described[79]. Chromosomes were stained with DAPI (2 µg/ml). Slides were observed using a Zeiss Axio Imager Z2 microscope. Images were acquired under a 100 × oil immersion objective, and processed with ZEN software. Immunolocalization performed on meiocytes with preserved three-dimensional structures was performed from 0.8–1.2 mm pistils (female meiocytes embedded in ovules) and anthers from 0.35–0.45 mm flower buds (male meiocytes), respectively[18,21]. Three primary antibodies were used for MLH1 foci number counting: anti-REC8 raised in rat[80] (laboratory code PAK036, dilution 1:250), anti-MLH1 raised in rabbit[81] (PAK017, 1:200), and anti-HEI10 raised in chicken[82] (PAK046, 1:5000). Secondary antibodies (dilution 1:250) were Abberior STAR ORANGE Goat anti-rat IgG (STORANGE-1007), STAR RED Goat anti-chicken IgY (STRED-1005) and STAR GREEN Goat anti-rabbit IgG (STGREEN-1002).Images were acquired under Laica THUNDER Imager system and Abberior instrument facility line (https://abberior-instruments.com/). For Leica THUNDER Imager system, 555-nm, 635-nm and 475-nm excitation lasers were used for STAR Orange, STAR Red and STAR GREEN, respectively. Images were deconvoluted using Large Volume Computational Clearing (LVCC) mode in a Laica LAS X 5.1.0 software. For Abberior instrument facility line, images were acquired with 561- and 640-nm excitation lasers (for STAR Orange and STAR Red, respectively) and a 775-nm STED depletion laser. Confocal images were taken with the same instrument with a 485-nm excitation laser (for STAR GREEN/ Alexa488). Images were deconvoluted by Huygens Essential (version 21.10, Scientific Volume Imaging, https://svi.nl/). Deconvoluted pictures were imported to Imaris x64 9.6.0 (https://imaris.oxinst.com/, Oxford Instruments, UK) for foci counting. The Spots module was used for MLH1 foci counting in diplotene and diakinesis cells. The vast majority of MLH foci colocalize with HEI10 foci (Table S1). Only double MLH1/HEI10 foci present on chromosomes were taken into account.

### Statistics & reproducibility

No statistical method was used to predetermine sample size. No data were excluded from the analyses. The experiments were randomized by genetic segregation of mutant and wild-type alleles in plant populations. The Investigators were not blinded to allocation during experiments and outcome assessment.

### Reporting summary

Further information on research design is available in the Nature Portfolio Reporting Summary linked to this article.

### Data availability

The list of identified COs in the female and male populations of wild type, HEI10[oe], *zyp1*, *recq4*, *zyp1* HEI10[oe], *recq4* HEI10[oe], *zyp1 recq4*, *zyp1 recq4* HEI10[oe] and *zyp1 recq4* HEI10[oe] *figl1* can be accessed in Supplementary Data 1. The number of COs, sequencing depth and source of each individual sample is provided in the Source data file of Fig. 1. The raw sequencing data generated in this study have been deposited in the ArrayExpress EMBL-EBI database under accession codes E-MTAB-14424, E-MTAB-14425, E-MTAB-14426, E-MTAB-14427, E-MTAB-14428 and E-MTAB-14430. Previously published datasets, deposited in the ArrayExpress EMBL-EBI database under accession codes E-MTAB-11254, E-MTAB-11696 and E-MTAB-9593, were also used in this study. Source data are provided with this paper.

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

## Acknowledgements

This work was supported by core funding from the Max Planck Society, and Alexander von Humboldt Fellowships to Q.L. and J.J. We thank Ian Henderson for kindly providing the HEI10 C2 line.

## Author contributions

R.M. lead the project. J.J. produced the genetic material, analyzed the cytology experiments and fertility (wild type, *zyp1*, *recq4*, *zyp1 recq4*, *zyp1* HEI10^oe^, *recq4* HEI10^oe^, *zyp1 recq4* HEI10^oe^ and *zyp1 mus81*). Q.L. analyzed the sequencing data and performed recombination, interference, and aneuploidy analyses. S.D. developed the protocol for female immunolocalization, produced the genetic material and analyzed the cytology experiments and fertility (*zyp1 recq4* HEI10^oe^ *figl1*). J.J., Q.L., and R.M. wrote the manuscript.

## Funding

## Competing interests

The authors declare the following competing interests: A Patent was granted to Institut National de la Recherge Agronomique (INRA) on the use of RECQ4 to manipulate meiotic recombination in crops, with RM listed among the inventors (US10,920,237/EP3149027). A patent was filed by the Max Planck Society on the combined use of RECQ4 and ZYP1 to manipulate recombination in crops, with R.M. J.J., Q.L., and S.D. listed as inventors (EP23179262. 14.06.2023).
