## [Peer Review file · Nature Communications]

Maximizing meiotic crossover rate reveals the map of Crossover Potential

Corresponding Author: Professor Raphael Mercier

Version 0:

Reviewer comments:

Reviewer #1

(Remarks to the Author)

The manuscript entitled "Maximizing meiotic crossover rate reveals the map of Crossover Potential" by Juli Jing and colleagues demonstrated an unprecedented elevation of meiotic crossovers in Arabidopsis by simultaneously mutating ZYP1 and RECQ4. The marginally affected fertility in *zyp1 recq4* opens new possibilities for plant breeding. Consistent observations of additional mutated crossover regulators in *zyp1 recq4* evidenced that the two pathways compete for a large but limited set of recombination intermediates. The authors termed the crossover landscapes of diverse mutants in both females and males as a single novel profile, namely Crossover Potential. I am particularly intrigued by the COp defined by the authors. Relating the density map of COp to chromatin markers in various mutants, for example, might be interesting. The experiments were well designed and the data are convincing. I do have a number of specific suggestions/questions about the manuscript for clarity.

Major :

- 1.Lines 86-87, "In Fig. 2n, the number of genetic CO expected to be observed if each MLH1 foci is converted into a CO is indicated by a dashed line (1 foci=0.5 CO)". Is there a paper in the literature that to estimate a frequency of CO with the method reported in the manuscript? And the explanation below Figure 2 suggests that "The dash line represents COs per gamete is on average one-half of COs per meiocyte." This expression is somewhat biased. Whether the number of MLH1/HEI10 foci per cell can stand for all COs per meiocyte. What about the foci of class II COs?
- 2.Lines 162-163:The authors showed that the *figl1* mutation led to a detectable decrease in recombination when combined with *zyp1 recq4 HEI10-oe*. Since it has been reported that RECQ4, FIGL1, and FANCM are three factors that limit crossovers (CO) and belong to three different pathways (Joiselle Blanche Fernandes et al. 2018. PNAS), I am wondering if it incorporating the *fancm* mutation into *zyp1 recq4 HEI10-oe* would further increase recombination as expected?
- 3.Lines 190-193: "...analyses based on genome coverage by sequencing did not detect any aneuploids in the progeny of *zyp1 recq4*." Can you please describe these analyses in more detail? I can't find a description of it in the methods section.
- 4.Lines 206-209: As described by the author, an increase in COs with mutated ZYP1 in certain contexts suggested an anti-class II CO function. Class II COs can be generated by at least two parallel pathways in Arabidopsis, which depend on either the structure-specific endonuclease (AtMUS81) or a homolog of Fanconi Anemia Complementation Group D2 (AtFANCD2). What about the phenotype of *zyp1 fancd2* during meiosis? I would like to hear more about whether abnormal chromosomal fragmentation would occur in *zyp1 fancd2*. On the other hand, would the *zyp1 fancd2 mus81* mutant exhibit more severely decreased fertility?
- 5.Both HEI10 levels and COI frequency correlate with SC lengths. How are the SC lengths affected by the CO landscape in *zyp1 recq4 HEI10oe* mutants?
- 6.Lines 103-105: My previous concern is still here that the number of MLH1 foci was unchanged in both female or male meiocytes in *zyp1* compared to wild-type. Previous studies in Arabidopsis demonstrated that the numbers of HEI10 foci were approximately twofold higher at diplotene in the *zyp1* mutants compared to wild type. The authors also stated that mutating ZYP1 in certain contexts (*recq4*, female HEI10oe) provokes an increase in COs but not MLH1 foci. However, what about the numbers of HEI10 foci in these mutants? Thus, whether ZYP1 also function in anti-class II CO should be carefully analyzed (Lines 206-208).
- 7.Lines 288-289: The method for building a machine-learning algorithm to predict the COP landscape should be clear, which should be included to the method section. To explore the predictive performance in the COP landscape of the model, could the authors predict what we would expect to find in other species, such as rice, wheat and yeast, whose chromatin state and sequence divergence have been published.
- 8.Could the authors comment in the Discussion on how important they predict the MSH4 gene to be in stabilizing meiosis in

allopolyploid *B. napus*, and how generalizable these results might be to other allopolyploids? To explore the predictive performance in the COP landscape of the model, could the authors predict what we would expect to find in other species, such as wheat, whose chromatin state and sequence divergence have been published.

9. The largest crossover increase was reached in *zyp1 recq4*, what about the genome-wide landscapes of DNA methylation level and heterochromatic histone modifications in the *zyp1 recq4* mutants. It has been reported that the recombination landscape is remodeled in DNA methylation mutants. However, whether the DNA methylation would be remodeled under the largest recombination frequency increase reached in *zyp1 recq4* is unknown.

Minor:

1. Figure 1: It would be better to perform correlation analysis between mean number of COs per transmitted chromatid and chromosome size.

2. Figure. 1b and 3c: I think it would be beneficial to show on the plot the statistical significance between male and female meiosis for each genotype.

3. Lines 150-152: The conclusion is that ZYP1 plays a role in preventing type II CO in addition to regulating type I CO. Was there any attempt to explore whether there are any interactions between ZYP1 and the COI and COII factors separately, which could better explain the function of ZYP1?

4. Figure 2b-m: MLH1-HEI10 co-foci were used in diplotene to indicate class I CO sites. Is there any basis for the choice? It might be good to add the references.

5. Figure 2p: The legend for histogram is incomplete, and only the yellow icon for Chromosome connection could be seen. Please check it.

6. MLH1-HEI10 co-foci were used in diplotene to indicate class I CO sites. Is there any basis for the choice? It might be good to add the references.

7. Figure 3a: The n-value of *zyp1 HEI10oe*, *recq4 HEI10oe* and *zyp1 recq4* is insufficient, please increase the number of samples to enhance reliability of data.

Reviewer #2

(Remarks to the Author)

Meiotic recombination is at the heart of plant breeding since it allows reshuffling of genes and alleles and contributes to the creation of new original gene combinations. However, recombination remains limited both in terms of quantity and localisation. In their study, the authors combined several mutants known to increase recombination in *Arabidopsis*. They showed that combining simultaneously mutants for *Atzip1* and *Atrecq4* gave the best increasing ever seen without affecting significantly the fertility. Interestingly, recombination was more increased in female than in male. Also, recombination rate seems to achieve a maximum in this double mutant. Adding other mutations (*HEI10oe* and *Atfig1*) did not increase further overall recombination rate suggesting an upper-limit of CO formation and a competition between class I and class II COs. Moreover, the recombination profiles in male and female became similar leading to a new Crossover Potential (COP) governed by intrinsic DNA features, the most important being SNP diversity.

Results were obtained using a set of eight single (3), double (3), triple (1) and quadruple (1) mutants for four genes (*AtZyp1*, *AtRecQ4*, *AtHei10*, *AtFig1*). Recombination events (crossovers) were counted in a huge number of BC1 individuals (3629) and completed with data from literature (for *Atrecq4/Atfig1* double mutant). Crossovers were extracted from sequencing data. Fluorescent in situ hybridization was also performed using MLH1 and HEI10 which also mark CO events, and common MLH1/HEI10 foci were also counted. In this case, up to 25 cells have been counted. However, are all cells extracted from the same plant or from different plants? If different plants, how many were used? For some mutants, you have an extremely huge variability between individuals (for *Atzyp1/Atrecq4*, range ~6-76 COs/gamete). On what bases was (or were) chosen the individual(s) for foci analyses?

L88-91: The MLH1 foci have been produced on male meiocytes but not on female meiocytes, I guess. Or this is not indicated in the manuscript when female meiocyte is used. How can you conclude regarding the class I/class II ratio in female?

Moreover, L100-102, since the increase of the number of COs is higher in female for *Atrecq4* mutants (which is involved in class II COs), this would suggest that class II COs are prevalent in female gametes? Please comment on that.

L94-95: For *HEI10oe*, the number of COs is lower in female because the initial number of COs was also lower. However, the increase (2.2- vs 1.9-fold) seems higher. Can you check if this difference is significant? The same is even more visible in *Atzyp1* (increase twice more in female). Check as well the significance.

L122-134: It is not clear whether foci are obtained on male meiocytes but in female BC1 or on female meiocytes derived from ovules? This is not indicated in Material & Methods. Also, in figure 2g-l, it seems that some foci are specific from MLH1 and others from HEI10. Does this mean that HEI10-specific foci are type II COs? What about MLH1-specific foci (there are some obvious on Figure 2i)? Are these resolved through a different pathway? Please comment on that.

L190-195: If you have 16% of meiotic cells with abnormal structure but no aneuploid lines, does this mean that deficient meiocytes lead to aborted pollen grains or does the seeds produced are unable to germinate and are therefore eliminated (what is the germination rate)? Can you elaborate on that?

L212-213: If ZYP1 prevents repair through MUS81 pathway, does this mean that it favours class I repair or another pathway?

L218-240: Figure 4 is extremely difficult to follow and we can only trust what the authors are saying. Maybe it would be better to have separate profiles in supplementary data and only zooms when regions are described in details?

L259-274: There is a strong difference between male in female in the middle of the right arm of chromosome 3 even when data are normalized (figure 6a-b). Is there any explanation for this?

L310-312: This is true in *Arabidopsis*. However, the fertility of some mutants can be strongly affected in other crops such as *Fig1* or *Fancm* mutants as this has been shown by the authors (see Mieulet et al. 2018; *Nat. Plants*

<https://doi.org/10.1038/s41477-018-0311-x>). Mutants for *RECQ4* also show some sterility in wheat (see Bazile et al. 2024 *Front. Plant Sci.* <https://doi.org/10.3389/fpls.2023.1342976>). So the usefulness in plant breeding must be toned down.

L359: The hypothesis that more DSBs are formed in female meiocytes can easily be checked using DMC1 foci. Since you counted MLH1/HEI10 foci (which is not clear) in female meiocytes, you could count DMC1 foci as well.

It is surprising that the number of NCOs in these mutants is never discussed. Since most data used here come from sequencing, is there a chance to see some NCO events? If yes, how many? Is there a difference between wildtype and mutants? Could you elaborate on that?

Based on the results, it seems that as soon as you mutate *Atrecq4*, the variability between individuals becomes much higher than with the other single mutants. Can you comment on that?

L403: Please, specify the number of reads generated and the coverage per individual this represents.

Nothing regarding statistics is given. Have statistics been applied on CO counts? Which statistics? Please, give details for that and apply statistics on every set of data to underline significance of your results. Mutant *Atmus81* is not mentioned. Are these already published data as it is referred to several publications?

Reviewer #3

(Remarks to the Author)

Meiotic recombination is a fundamental basis for molecular breeding in crops, but the current understanding in manipulation of the maximum increment of meiotic crossovers in plants are very limited. In this manuscript, Jing et al simultaneously disrupted multiple anti-crossover factors in *Arabidopsis* and evaluated the CO rate in genome wide scale. They reported several observations which are really intriguing. First, they showed that mutation of both *ZYP1* and *RECQ4* lead to the currently maximum number of COs with ~70 in female and ~48 in male. Second, the triple mutants with combination of *zyp1 recq4 HEI10oe* does not further increase the total CO number but favors more class I COs at the expense of class II COs. Third, heterochiasmy has been inverted in the hyper-recombinant mutants. Finally, they termed Crossover Potential (COP) on the basis of analyses of almost 50000 COs and proposed that COP can be predicted according to the sequence divergence and chromatin features. Nevertheless, each one observation deserves further investigation to fully understand the underlying mechanism. Overall, it's a well-knitted manuscript with rigorous experimental design and comprehensive evidence. I have only a few questions which should be considered by the authors to further improve the manuscript.

Specific comments:

1. I do agree that the current observation of COs in the *zyp1 recq4* mutant background is the maximum number, but I am not sure whether it could be the real maximum. I just think about if we disrupt other factors related to chromatin modification in combination of the anti-CO factors, whether it could have an effect on increasing CO further. Since the other species with relatively simple chromatin modification have more COs.

2. Since the most visible evidence to support the CO pathways is reflected by the observation of MLH1 foci. In Figure 2, only the pachytene stage chromosomes were shown, if the author can provide the diakinesis stage which will be very helpful.

3. The fertility in multiple mutants background were only estimated by the counting the seeds in silique. I favor to the number of viable pollen grains and tetrad. Because there are ~16% of abnormal metaphase I cells in *zyp1 recq4*, which will be amplified further after the second division of meiosis. If the authors also have these data. Please include.

4. The results of percentage of aneuploidy in Figure 3C is theoretically not matched to data in Figure 3d-3o. Again, the chromosome morphology in *zyp1 HEI10oe* was not included.

5. *HEI10oe* showed more frequent chromosome connections and worse fertility. Line 197 you explained: "possibly due to a chromosomal rearrangement associated with the transgene in the *HEI10oe* C2 line." But how could that affects the segregation so much? I'm wondering if you choose a better *HEI10oe* line, could you reach an even higher CO rate?

6. Line 194, it was said "suggestive of unrepaired recombination intermediates, which might cause reduced fertility in *zyp1 recq4*". Please explain what's the meaning of unrepaired recombination intermediates. It could abnormal recombination intermediates.

7. It's fascinating that under *zyp1* mutant background, heterochiasmy of class I CO is abolished, while the inverted heterochiasmy of class II CO has been established. This raised a question that *zyp1 recq4*♀ and *zyp1 recq4 HEI10oe*♀ have similar female COs with 70 and 68 respectively, why *zyp1 recq4*♂ and *zyp1 recq4 HEI10oe*♂ have male COs with 46 and 52 respectively. I am curious about why COP cannot assure the similar number of COs between them in male.

8. Why you ignore Chr4 starting from Fig 5? Is it because of an inversion there?

9. One of the explanations to increase the CO in female more than that in male in corresponding mutants is to increase more DSBs. Is it possible to provide evidence to support this?

Reviewer #4

(Remarks to the Author)

This paper is very interesting and elegant, and important contribution to our understanding of diversity generation in plants. It is very straightforward and well-presented; I have few comments, none of them major, only to suggest small enhancements.

The introduction is very nice until it abruptly ends with little context or summary. Some development of the end of the introduction (e.g. a small para summarising) would be helpful for flow and additional reinforcement of context for the nonexpert.

One perhaps overly obvious thing is never explicitly said that I can see: were all samples individually sequenced or is this somehow instead based off sequencing pools? The former seems unsaid (obvious), but if so, more details on the method would be helpful. As it is now the methods only says (Ln 400) 'population'... 'were sequenced'. Obviously as stated these are nonrepeatable methods and should be checked overall for sufficient detail.

17 Why should fertility be strongly affected by higher COs in a homozygous selfing diploid? I realise this is clear to the expert but not to the non-expert, which is important in this generalist journal.

This comes up again later given that SNP density (extremely low in A.th) is of 'the utmost importance' (line 291) for explaining variation in the model developed. This should be discussed after the model in terms of effects (limitations) of this (ultimately rather strange, in terms of plant diversity) model system.

25 That the 'COp' can be 'accurately predicted using only sequence divergence and chromatin features' is in theory very exciting, I'm less convinced in its impact here given the system, being highly homozygous and lacking diversity. Some word on possible limitations (if any) would be welcome.

87 Maybe it's against convention in the field, but should not the singular of 'foci' be 'focus'?

141 Suggest changing 'the so-far champion' to the 'the champion to date'

181 it's clear it's the case but it would be good to frame the motivation for fertility assessment. I.e. why is this a valid way to look at the possible impact of extreme CO? Esp in low-diversity A.th?

It would be very good to have some mention in the main text what your ability to detect CO in different genomic feature, esp centromeres. Also, is the reference masked? What is the possible effect of multiple mapping and filtering of short reads in repeat regions, esp centromeres and flanking regions? How could this affect determination of COs in these regions (if at all, I understand, but it should be clear in the main text what possible sources of ascertainment bias are and how they may or may not be an issue in the analysis. A small paragraph on this would be a valuable addition, I think.

Version 1:

Reviewer comments:

Reviewer #1

(Remarks to the Author)

The authors have adequately addressed my comments regarding the manuscript. I was satisfied with the scientific meaning and content of the revision. I have no further comments for the revised manuscript.

Reviewer #2

(Remarks to the Author)

This paper is a revised version of a previous submission of the manuscript entitled "Maximizing meiotic crossover rate reveals the map of crossover potential".

This new version has been deeply modified according the comments that reviewers emitted. This has greatly contributed to improve its quality. The answers are extremely well detailed and are convincing. When additional experiments were suggested, these have been done or justification of why this could not be done is elaborated. All conclusions and claims are supported and no flaws are identified. This is an extremely well designed work that will undoubtedly contribute to improve knowledge in the field of recombination in plants.

Reviewer #3

(Remarks to the Author)

I appreciate the authors who have done a great job to address my concerns. I have no further concerns on the revised manuscript and look forward to seeing the paper published in the Nature Communications.

Reviewer #4

(Remarks to the Author)

The revision has satisfied my concerns.

REVIEWER COMMENTS

Reviewer #1 (Remarks to the Author):

The manuscript entitled "Maximizing meiotic crossover rate reveals the map of Crossover Potential" by Juli Jing and colleagues demonstrated an unprecedented elevation of meiotic crossovers in Arabidopsis by simultaneously mutating ZYP1 and RECQ4. The marginally affected fertility in *zyp1 recq4* opens new possibilities for plant breeding. Consistent observations of additional mutated crossover regulators in *zyp1 recq4* evidenced that the two pathways compete for a large but limited set of recombination intermediates. The authors termed the crossover landscapes of diverse mutants in both females and males as a single novel profile, namely Crossover Potential. I am particularly intrigued by the CO_p defined by the authors. Relating the density map of CO_p to chromatin markers in various mutants, for example, might be interesting. The experiments were well designed and the data are convincing. I do have a number of specific suggestions/questions about the manuscript for clarity.

> Thank you for your review and your positive assessment of the work.

Major :

1.Lines 86-87, "In Fig. 2n, the number of genetic CO expected to be observed if each MLH1 foci is converted into a CO is indicated by a dashed line (1 foci=0.5 CO) ". Is there a paper in the literature that to estimate a frequency of CO with the method reported in the manuscript? And the explanation below Figure 2 suggests that "The dash line represents COs per gamete is on average one-half of COs per meiocyte." This expression is somewhat biased. Whether the number of MLH1/HEI10 foci per cell can stand for all COs per meiocyte. What about the foci of class II COs?

> Indeed, this was ambiguous, and we have rewritten the paragraph for clarity:

"Note that the mean number of CO per gamete is expected to be half the number of cytological CO (or chiasma). This is because a chiasma affects only two of the four chromatids of a chromosome pair, and a gamete inherits a single chromatid (see figure S11 in ²⁰). Thus, under the assumption that each MLH1 focus is converted into a CO and that all chiasmata are marked by MLH1, the frequency of genetic CO per gamete/BC1 is expected to be half of the frequency of MLH1 foci per cell (1 focus=0.5 CO, indicated by a dashed line in Fig. 2n). If a proportion of chiasmata/CO are not marked by MLH1 (i.e. class II COs), the frequency of genetic CO per gamete is expected to be higher than half of the frequency of MLH1 foci per cell (an upward deviation from the dashed line in Fig. 2n)."

We edited the legend of Figure 2 accordingly.

2.Lines 162-163:The authors showed that the *figl1* mutation led to a detectable decrease in recombination when combined with *zyp1 recq4 HEI10-oe*. Since it has been reported that RECQ4, FIGL1, and FANCM are three factors that limit crossovers (CO) and belong to three different pathways (Joiselle Blanche Fernandes et al. 2018. PNAS), I am wondering if it incorporating the *fancm* mutation into *zyp1 recq4 HEI10-oe* would further increase recombination as expected?

> This is a heavy experiment that would need multiple generations of crosses, that we consider beyond the scope of the study. It should also be noted that *fancm* has little effect on CO in the *col/ler* hybrid background, and notably does not further increase CO when combined with *recq4* (figure 3A in Fernandes et al 2018). It is thus likely that muting *fancm* in *zyp1 recq4 HEI10-oe* would not further increase COs.

3. Lines 190-193: "...analyses based on genome coverage by sequencing did not detect any aneuploids in the progeny of *zyp1 recq4*." Can you please describe these analyses in more detail? I can't find a description of it in the methods section.

> The method used to detect aneuploids is described in lines 521-526 of the revised manuscript.

4.Lines 206-209: As described by the author, an increase in COs with mutated ZYP1 in certain contexts suggested an anti-class II CO function. Class II COs can be generated by at least two parallel pathways in Arabidopsis, which depend on either the structure-specific endonuclease (AtMUS81) or a homolog of Fanconi Anemia Complementation Group D2 (AtFANCD2). What about the phenotype of *zyp1 fancd2* during meiosis? I would like to hear more about whether abnormal chromosomal fragmentation would occur in *zyp1 fancd2*. On the other hand, would the *zyp1 fancd2 mus81* mutant exhibit more severely decreased fertility?

> These are interesting suggestions but we consider these experiments beyond the scope of this study and should be included in future work.

5.Both HEI10 levels and COI frequency correlate with SC lengths. How are the SC lengths affected by the CO landscape in *zyp1 recq4 HEI10oe* mutants?

> Unfortunately, we did not image pachytene cells when we analyzed the *zyp1 recq4 HEI10oe col/ler* hybrid plants. However, SC length is not affected in *zyp1, HEI10oe, and zyp1 HEI10oe*, despite a large increase in class I COs (Durand et al, 2022). We thus do not expect SC length to be modified in *zyp1 recq4 HEI10oe*.

6.Lines 103-105: My previous concern is still here that the number of MLH1 foci was unchanged in both female or male meiocytes in *zyp1* compared to wild-type. Previous studies in Arabidopsis demonstrated that the numbers of HEI10 foci were approximately twofold higher at diplotene in the *zyp1* mutants compared to wild type. The authors also stated that mutating ZYP1 in certain contexts (*recq4*, female *HEI10oe*) provokes an increase in COs but not MLH1 foci. However, what about the numbers of HEI10 foci in these mutants? Thus, whether ZYP1 also function in anti-class II CO should be carefully analyzed (Lines 206-208).

> The sentence in line 103-105 was ambiguous and is modified in the revised manuscript. The number of MLH1 foci is increased in both female and male meiocytes compared to wild-type (Figure 2a female, from 6.8 in wild-type to 12.2 in *zyp1*. Male, from 10.3 in wild-type to 12.7 in *zyp1*), and the numbers are coherent with previous data also performed in *col/ler* hybrids (Durand et al, 2022). The absolute numbers are slightly different from other HEI10/MLH1 published data produced in pure Col (Capilla et al 2020, France et al 2020, Durand et al

2022), which is not unexpected as the frequency of CO differs slightly between Col and Col/Ler (Lian et al, 2022).

We used HEI10/MLH1 co-foci as a reliable marker of CO sites. In all the genotypes, MLH1 and HEI10 foci at diplotene/diakinesis almost perfectly match (on average 97% of MLH1 foci colocalize with HEI10 foci, and 98% of HEI10 foci colocalize with MLH1). We have added this data to the manuscript (table S1 for colocalization frequencies and raw data in the Source data file of Figure 2). The small percentages of non-matching foci are possibly due to technical reasons (e.g. threshold effect)

7.Lines 288-289: The method for building a machine-learning algorithm to predict the COP landscape should be clear, which should be included to the method section. To explore the predictive performance in the COP landscape of the model, could the authors predict what we would expect to find in other species, such as rice, wheat and yeast, whose chromatin state and sequence divergence have been published.

> We added the details in the Methods section. To date, we don't have COP data in other species than Arabidopsis to train a model and to compare observations with predictions. Such analysis is thus beyond the scope of the current manuscript.

8.The largest crossover increase was reached in *zyp1 recq4*, what about the genome-wide landscapes of DNA methylation level and heterochromatic histone modifications in the *zyp1 recq4* mutants. It has been reported that the recombination landscape is remodeled in DNA methylation mutants. However, whether the DNA methylation would be remodeled under the largest recombination frequency increase reached in *zyp1 recq4* is unknown.

> This is an interesting suggestion, but to test if the modified recombination landscape affects chromatin, the methylation and histone modification analyses should be done on meiotic cells. In Arabidopsis, this is very challenging.

Minor:

1.Figure 1: It would be better to perform correlation analysis between mean number of COs per transmitted chromatid and chromosome size.

> We have added this analysis as Figure S5.

2. Figure. 1b and 3c: I think it would be beneficial to show on the plot the statistical significance between male and female meiosis for each genotype.

> Yes. This is included in the revised manuscript.

3.Lines 150-152: The conclusion is that ZYP1 plays a role in preventing type II CO in addition to regulating type I CO. Was there any attempt to explore whether there are any interactions between ZYP1 and the COI and COII factors separately, which could better explain the function of ZYP1?

> The *zyp1* mutation alone seems to increase only class I COs (figure 2n). This is consistent with the very low level of CO observed in *zyp1 msh5* double mutant (Capilla et al, 2020). Thus the anti-class II activity of *zyp1* is only revealed when combined with other anti-co factors (e.g. *recq4*).

4. Figure 2b-m: MLH1-HEI10 co-foci were used in diplotene to indicate class I CO sites. Is there any basis for the choice? It might be good to add the references.

> The number of MLH1 foci increases during pachytene and from pachytene to diplotene and then is stable from diplo to diakinesis (Chelysheva 2020, Duroc 214, Mathilde Grelon, Personal communication). The number of Foci at diplotene/diakinesis thus represents the final number of foci/class I COs). We have added the references in the revised manuscript).

5. Figure 3p: The legend for histogram is incomplete, and only the yellow icon for Chromosome connection could be seen. Please check it.

> Thank you for pointing this out. This is corrected in the revised manuscript.

6. MLH1-HEI10 co-foci were used in diplotene to indicate class I CO sites. Is there any basis for the choice? It might be good to add the references.

> See comment 4.

7. Figure 3a: The n-value of *zyp1 HEI10oe*, *recq4 HEI10oe* and *zyp1 recq4* is insufficient, please increase the number of samples to enhance reliability of data.

> This number was limited by the number of plants in the segregating population. Each point is one plant, and 10 fruits were counted for each plant, given a total of 60 fruits for *zyp1 HEI10oe*, 30 fruits for *recq4 HEI10oe*, and 80 for *zyp1 recq4*, which we consider sufficient for a reliable assessment of the fertility. This is clarified in the legend of the revised manuscript.

Reviewer #2 (Remarks to the Author):

Meiotic recombination is at the heart of plant breeding since it allows reshuffling of genes and alleles and contributes to the creation of new original gene combinations. However, recombination remains limited both in terms of quantity and localisation. In their study, the authors combined several mutants known to increase recombination in Arabidopsis. They showed that combining simultaneously mutants for *Atzip1* and *Atrecq4* gave the best increasing ever seen without affecting significantly the fertility. Interestingly, recombination was more increased in female than in male. Also, recombination rate seems to achieve a maximum in this double mutant. Adding other mutations (*HEI10oe* and *Atfig11*) did not increase further overall recombination rate suggesting an upper-limit of CO formation and a

competition between class I and class II COs. Moreover, the recombination profiles in male and female became similar leading to a new Crossover Potential (COP) governed by intrinsic DNA features, the most important being SNP diversity.

Results were obtained using a set of eight single (3), double (3), triple (1) and quadruple (1) mutants for four genes (*AtZyp1*, *AtRecQ4*, *AtHei10*, *AtFig11*). Recombination events (crossovers) were counted in a huge number of BC1 individuals (3629) and completed with data from literature (for *Atrecq4/Atfig11* double mutant). Crossovers were extracted from sequencing data.

Fluorescent in situ hybridization was also performed using MLH1 and HEI10 which also mark CO events, and common MLH1/HEI10 foci were also counted. In this case, up to 25 cells have been counted. However, are all cells extracted from the same plant or from different plants? If different plants, how many were used? For some mutants, you have an extremely huge variability between individuals (for *Atzyp1/Atrecq4*, range ~6-76 COs/gamete). On what bases was (or were) chosen the individual(s) for foci analyses? L88-91: The MLH1 foci have been produced on male meiocytes but not on female meiocytes, I guess. Or this is not indicated in the manuscript when female meiocyte is used. How can you conclude regarding the class I/class II ratio in female?

> Thank you for your evaluation of the work and your suggestions for improvements.

MLH1/HEI10 foci were indeed counted in both male and female meiocytes of F1 hybrids (Col/Ler), extracted from anthers and pistils, respectively.

One or two F1 plants were used per genotype:

Wild-type female: 1 plant(ID: 220129-103); male: 2 plants(220129-120, 220129-103).

zyp1 female: 1 plant(220591-325); male: 1 plant(220591-325).

HEI10oe female: 1 plant(230001-59); male: 1 plant(230001-59).

recq4 female: 1 plant(220745-14); male: 1 plant(220745-14).

zyp1 HEI10oe female: 2 plants (230207-76, 230207-95); male: 2 plants(230207-76, 230207-95).

Recq4 HEI10oe female: 2 plants(220591-345, 220591-32); male: 2 plants(220591-345, 220591-32)

zyp1 recq4 female: 2 plants(220129-20, 220129-45); male: 2 plants(220129-20, 220129-45)

zyp1 recq4 HEI10oe female: 2 plants(230001-59, 230001-66); male: 2 plants(230001-66, 230271-110)

zyp1 recq4 HEI10oe fig11 male: 2 plants(230271-90, 230271-98).

For full transparency, the identity of individual plants used for foci counting was added to the File “source_data_Figure2”, and this is indicated in the figure legend

F1 hybrids were crossed as female and as male with Col wild-type (Figure S1), to generate two populations of BC1 seeds (“female” and “male”) . B1 plantlets were sequenced to detect CO that occurred in the female and male meiocytes of the F1 hybrid, respectively. We did not apply any selection on the BC1 seeds/plantlets to be sequenced.

Thus, we were able to compare the MLH1/HEI10 foci in female and male meiocytes in F1 hybrids to the genetic CO produced in these hybrids and transmitted to the BC1 populations, and thus infer class I/class II ratios in both female and males. We observed indeed huge variability between gametes (BC1 individuals) in some genotypes, which indicated a massive deregulation of the meiotic machinery.

We edited the manuscript to clarify all these points.

Moreover, L100-102, since the increase of the number of COs is higher in female for *Atrecq4* mutants (which is involved in class II COs), this would suggest that class II COs are prevalent in female gametes? Please comment on that.

> Yes, this is exactly what we conclude later in the manuscript (line 356-357 of the original manuscript). In all mutant contexts, female meiosis is more prone to make class II CO than male meiosis. We edited the manuscript to clarify this point at both locations in the text.

L94-95: For *HEI10oe*, the number of COs is lower in female because the initial number of COs was also lower. However, the increase (2.2- vs 1.9-fold) seems higher. Can you check if this difference is significant?

> We performed a test and, indeed, the fold increases are significantly different. This is added in the revised manuscript.

The same is even more visible in *Atzyp1* (increase twice more in female). Check as well the significance.

> As expected, the difference in fold increase is very significant. Also added to the text.

L122-134: It is not clear whether foci are obtained on male meiocytes but in female BC1 or on female meiocytes derived from ovules? This is not indicated in Material & Methods.

> The foci are obtained from the female meiocytes (from ovules) of the F1. We clarified this both in the results and the Material & Methods sections of the revised manuscript.

Also, in figure 2g-I, it seems that some foci are specific from MLH1 and others from HEI10. Does this mean that HEI10-specific foci are type II COs? What about MLH1-specific foci (there are some obvious on Figure 2i)? Are these resolved through a different pathway? Please comment on that.

> In all the genotypes, MLH1 and HEI10 foci at diplotene/diakinesis almost perfectly match (on average 97% of MLH1 foci colocalize with HEI10 foci, and 98% of HEI10 foci colocalize with MLH1). We have added this data to the manuscript (table S1 for colocalization frequencies and raw data in the Source data file of Figure 2). The small percentages of non-matching foci are possibly due to technical reasons (e.g. threshold effect). Figure 2g shows, inevitably, the projection of the 3D images, which could make some foci appear fainter. The scoring of the foci was performed on the complete stack. We have thus included the full Z-stack of the corresponding images to figures 2g-I in the sup data.

L190-195: If you have 16% of meiotic cells with abnormal structure but no aneuploid lines, does this mean that deficient meiocytes lead to aborted pollen grains or does the seeds

produced are unable to germinate and are therefore eliminated (what is the germination rate?)? Can you elaborate on that?

> The number of seeds per fruit is reduced by ~25% in *zyp1 recq4* compared to wild-type (figure 3a). In Arabidopsis, there is ~60-70 ovules per developing fruit, each containing one meiocyte that produces one spore and ultimately one seed. The number of male gametes (in the pollen grain) is in very large excess. The meiotic defects in *zyp1 recq4* are thus likely the cause of the reduced seed number (inviability of female gamete or young embryo).

L212-213: If ZYP1 prevents repair through MUS81 pathway, does this mean that it favours class I repair or another pathway?

> We discussed this point in the revised manuscript:

“This points to a function of ZYP1 in preventing class II COs, which is revealed only when RECQ4 is absent, in addition to its documented function in regulating class I COs^{17,18}. It suggests that recombination precursors (DNA double-strand breaks) could be increased in the absence of ZYP1, as shown when the homologous protein Zip1 is defective in yeast^{44,45})

L218-240: Figure 4 is extremely difficult to follow and we can only trust what the authors are saying. Maybe it would be better to have separate profiles in supplementary data and only zooms when regions are described in details?

> Figure 4 is indeed very dense, but we believe it is important to show the global profiles and to avoid cherry-picking. To help the reader look into any detail, we added a series of supplementary figures to show each chromosome on a full page (Figure S6).

L259-274: There is a strong difference between male and female in the middle of the right arm of chromosome 3 even when data are normalized (figure 6a-b). Is there any explanation for this?

> Indeed, there is a difference between males and females. It looks to be driven by *zyp1 recq4* HEI10oe, *recq4* HEI10oe and *zyp1* HEI10oe. We have to admit that we have no good explanation for this. We are unaware of any rearrangement or genomic feature that would make this region special.

L310-312: This is true in Arabidopsis. However, the fertility of some mutants can be strongly affected in other crops such as Fig11 or Fancm mutants as this has been shown by the authors (see Mieulet et al. 2018; Nat. Plants <https://doi.org/10.1038/s41477-018-0311-x>). Mutants for RECQ4 also show some sterility in wheat (see Bazile et al. 2024 Front. Plant Sci. <https://doi.org/10.3389/fpls.2023.1342976>). So the usefulness in plant breeding must be toned down.

> We have included these considerations and tone down the application potential in the revised manuscript.

L359: The hypothesis that more DSBs are formed in female meiocytes can easily be checked using DMC1 foci. Since you counted MLH1/HEI10 foci (which is not clear) in female meiocytes, you could count DMC1 foci as well.

> This is a good suggestion, that we have explored. However, DMC1 immunolocalization is more delicate than MLH1/HEI10 immunolocalization and female meiocytes are less accessible than male meiocytes. Altogether, and despite our efforts, we have not succeeded in producing satisfactory images of DMC1 on female meiocytes. In addition, it should be noted that DMC1 is not a perfect proxy for DSB frequency, as a difference in DMC1 foci counts can be due also to a different turnover of the foci/repair time.

It is surprising that the number of NCOs in these mutants is never discussed. Since most data used here come from sequencing, is there a chance to see some NCO events? If yes, how many? Is there a difference between wildtype and mutants? Could you elaborate on that?

> NCOs are notoriously difficult to detect in Arabidopsis. Indeed, they are rare (~1-5 NCO gene conversion per meiosis) and very short (most of them converting a single SNPs). Thus, detecting NCOs confidently and sorting them from sequencing errors need super-deep sequencing, and preferentially sequencing of the four tetrads (Wijnker E, et al The genomic landscape of meiotic crossovers and gene conversions in Arabidopsis thaliana. Elife. 2013 Dec 17;2:e01426. doi: 10.7554/eLife.01426). The 2-5X sequencing used in this study does not allow the identification of NCOs. A preprint by the Mezard group, using deep sequencing of tetrads, detected an increase of NCOs in *figl1*, but not in *recq4* (Charif et al. Biorxiv 2024 <https://doi.org/10.1101/2024.09.30.615791>).

Based on the results, it seems that as soon as you mutate *Atrecq4*, the variability between individuals becomes much higher than with the other single mutants. Can you comment on that?

> The *recq4* mutation boosts class II COs, which are not subjected to interference. One of the consequences of interference is to constrain the number of CO per chromosome, and thus reduce the variability of the number of COs per cell. Consequently, the variability between gametes is much higher. We added a comment in the revised manuscript.

L403: Please, specify the number of reads generated and the coverage per individual this represents.

Nothing regarding statistics is given. Have statistics been applied on CO counts? Which statistics? Please, give details for that and apply statistics on every set of data to underline significance of your results. Mutant *Atmus81* is not mentioned. Are these already published data as it is referred to several publications?

> We have added in the supporting data a file with the number of reads, coverage, and number of CO in each sample (Source data figure 1). We included in our analysis samples for previous populations to increase the power (e.g. wild-type samples from this study, plus wild-type from previous experiments in the lab). Tests on CO counts were applied (Mann Whitney test, Figure 1, and throughout the text). We have also added the details of the *mus81* mutant in the methods.

Reviewer #3 (Remarks to the Author):

Meiotic recombination is a fundamental basis for molecular breeding in crops, but the current understanding in manipulation of the maximum increment of meiotic crossovers in plants are very limited. In this manuscript, Jing et al simultaneously disrupted multiple anti-crossover factors in Arabidopsis and evaluated the CO rate in genome wide scale. They reported several observations which are really intriguing. First, they showed that mutation of both ZYP1 and RECQ4 lead to the currently maximum number of COs with ~70 in female and ~48 in male. Second, the triple mutants with combination of *zyp1 recq4 HEI10oe* does not further increase the total CO number but favors more class I COs at the expense of class II COs. Third, heterochiasmy has been inverted in the hyper-recombinant mutants. Finally, they termed Crossover Potential (COP) on the basis of analyses of almost 50000 COs and proposed that COP can be predicted according to the sequence divergence and chromatin features. Nevertheless, each one observation deserves further investigation to fully understand the underlying mechanism. Overall, it's a well-knitted manuscript with rigorous experimental design and comprehensive evidence. I have only a few questions which should be considered by the authors to further improve the manuscript.

> Thank you for your positive evaluation of the work.

Specific comments:

1. I do agree that the current observation of COs in the *zyp1 recq4* mutant background is the maximum number, but I am not sure whether it could be the real maximum. I just think about if we disrupt other factors related to chromatin modification in combination of the anti-CO factors, whether it could have an effect on increasing CO further. Since the other species with relatively simple chromatin modification have more COs.

> True. We added this possibility in the discussion of the revised manuscript.

2. Since the most visible evidence to support the CO pathways is reflected by the observation of MLH1 foci. In Figure 2, only the pachytene stage chromosomes were shown, if the author can provide the diakinesis stage which will be very helpful.

> Figure two shows diplotene stages (desynapsed, but not condensed yet). Unfortunately, we do not have diakineses images for these mutants. We'd like also to mention that with this immunolocalization technique that preserves the 3D structure of the cells and chromosomes, the chiasmata are not easy to see, in contrast to techniques based on 3:1 ethanol:acetic acid fixation and spreading.

3. The fertility in multiple mutants background were only estimated by the counting the seeds in silique. I favor to the number of viable pollen grains and tetrad. Because there are ~16% of

abnormal metaphase I cells in *zyp1 recq4*, which will be amplified further after the second division of meiosis. If the authors also have these data. Please include.

> We have not produced these data. However, we do detect a clear reduction of fertility in *zyp1 recq4* by seed counting, which integrates both male and female fertility.

4. The results of percentage of aneuploidy in Figure 3C is theoretically not matched to data in Figure 3d-3o. Again, the chromosome morphology in *zyp1 HEI10oe* was not included.

The morphology of *zyp1 HEI10* chromosomes is described in Durand et al. 2022, showing the presence of univalents (also in single *zyp1*), likely at the origin of aneuploids. It is likely that chromosome fragmentations lead to the non-viability of the spores and thus not aneuploidy in the progeny. Please note also that the number of aneuploids in Figure C is small (ranging from 0 to 6), and should be over-interpreted.

5. *HEI10oe* showed more frequent chromosome connections and worse fertility. Line 197 you explained: “possibly due to a chromosomal rearrangement associated with the transgene in the *HEI10oe* C2 line.” But how could that affect the segregation so much? I’m wondering if you choose a better *HEI10oe* line, could you reach an even higher CO rate?

> We’d love to have a better *HEI10oe*, but we don’t at the moment. The C2 line is the most characterized in the community working on *Arabidopsis* meiosis. We suspect that the translocation (which is heterozygous in the hybrid context) leads to the formation of spores with an incomplete genome.

6. Line 194, it was said “suggestive of unrepaired recombination intermediates, which might cause reduced fertility in *zyp1 recq4*”. Please explain what’s the meaning of unrepaired recombination intermediates. It could be abnormal recombination intermediates.

> We replaced “unrepaired” by “abnormal”

7. It’s fascinating that under *zyp1* mutant background, heterochiasmy of class I CO is abolished, while the inverted heterochiasmy of class II CO has been established. This raised a question that *zyp1 recq4*♀ and *zyp1 recq4 HEI10oe*♀ have similar female COs with 70 and 68 respectively, why *zyp1 recq4*♂ and *zyp1 recq4 HEI10oe* ♂ have male COs with 46 and 52 respectively. I am curious about why COP cannot assure the similar number of COs between them in male.

> *zyp1 recq4* male and *zyp1 recq4 HEI10oe* male gametes have respectively an average of 23.5 and 25.6 CO, which is not significantly different from each other (Mann Whitney, $p=0.06$). We have added the test in the revised manuscript.

8. Why do you ignore Chr4 starting from Fig 5? Is it because of an inversion there?

Yes, it is because of the chromosome rearrangement, which is present in some genotypes.

9. One of the explanations to increase the CO in female more than that in male in

corresponding mutants is to increase more DSBs. Is it possible to provide evidence to support this?

> Yes, we agree that this is a good hypothesis. We have explored DMC1 immunolocalization as a potential proxy for DSB frequency. However, DMC1 immunolocalization is more delicate than MLH1/HE10 immunolocalization and female meiocytes are less accessible than male meiocytes. Altogether, and despite our efforts, we have not succeeded in producing satisfactory images of DMC1 on female meiocytes. In addition, it should be noted that DMC1 is not a perfect proxy for DSB frequency, as a difference in DMC1 foci counts can be due also to a different turnover of the foci/repair time.

Reviewer #4 (Remarks to the Author):

This paper is very interesting and elegant, and important contribution to our understanding of diversity generation in plants. It is very straightforward and well-presented; I have few comments, none of them major, only to suggest small enhancements.

The introduction is very nice until it abruptly ends with little context or summary. Some development of the end of the introduction (e.g. a small para summarising) would be helpful for flow and additional reinforcement of context for the nonexpert.

> The senior author of this study is not a fan of this common practice, with the argument that it is somehow redundant with the abstract. However, as suggested by the reviewer and previously supported by other authors of the manuscript, we included a small paragraph at the end of the introduction.

One perhaps overly obvious thing is never explicitly said that I can see: were all samples individually sequenced or is this somehow instead based off sequencing pools? The former seems unsaid (obvious), but if so, more details on the method would be helpful. As it is now the methods only says (Ln 400) 'population'... 'were sequenced'. Obviously as stated these are nonrepeatable methods and should be checked overall for sufficient detail.

> We have added more details in the material and methods section.

17 Why should fertility be strongly affected by higher COs in a homozygous selfing diploid? I realise this is clear to the expert but not to the non-expert, which is important in this generalist journal.

> In most eukaryotes, CO frequencies are limited to 1-3 per chromosome, leading to the idea that a higher frequency of CO could be detrimental, possibly disturbing chromosome segregation during meiotic divisions and fertility. We have included this in the revised manuscript.

This comes up again later given that SNP density (extremely low in A.th) is of ‘the utmost importance’ (line 291) for explaining variation in the model developed. This should be discussed after the model in terms of effects (limitations) of this (ultimately rather strange, in terms of plant diversity) model system.

> We have rephrased “ulmost importance”. Using SNP probably also captures other genomic features, as it is correlated with e.g. gene density.

25 That the ‘COp’ can be ‘accurately predicted using only sequence divergence and chromatin features’ is in theory very exciting, I’m less convinced in its impact here given the system, being highly homozygous and lacking diversity. Some word on possible limitations (if any) would be welcome.

> We have added a comment on this point in the revised manuscript.

87 Maybe it’s against convention in the field, but should not the singular of ‘foci’ be ‘focus’?

> Yes. Corrected.

141 Suggest changing ‘the so-far champion’ to the ‘the champion to date’

> ok. Done

181 it’s clear it’s the case but it would be good to frame the motivation for fertility assessment. I.e. why is this a valid way to look at the possible impact of extreme CO? Esp in low-diversity A.th?

>> In most eukaryotes, CO frequencies are limited to 1-3 per chromosome, leading to the idea that a higher frequency of CO could be detrimental, possibly disturbing chromosome segregation during meiotic divisions and fertility. We have included this in the revised manuscript.

It would be very good to have some mention in the main text what your ability to detect CO in different genomic feature, esp centromeres. Also, is the reference masked? What is the possible effect of multiple mapping and filtering of short reads in repeat regions, esp centromeres and flanking regions? How could this affect determination of COs in these regions (if at all, I understand, but it should be clear in the main text what possible sources of ascertainment bias are and how they may or may not be an issue in the analysis. A small paragraph on this would be a valuable addition, I think.

> Lack of markers locally (e.g. centromeres) does not prevent the detection of CO (because we can use flanking larkers), but prevents determining their position. We added a paragraph in the main to explain the limitations of the method

“This method allows CO measurement with high accuracy, with some limitations. First, the CO position cannot be determined in regions lacking reliable markers (e.g. centromeric regions); but note that the presence/absence of CO in such areas can be reliably assessed by analyzing flanking markers. Second, with our stringent parameters, double-COs less than

90kb apart, if they exist, and terminal CO less than 30kb away from the telomere would be missed (see methods). “